# PolaFormer: Polarity-aware Linear Attention for Vision Transformers

**Weikang Meng**[1,2]**, Yadan Luo**[3]**, Xin Li**[2]**, Dongmei Jiang**[2]**, Zheng Zhang**[1*]

[1] Harbin Institute of Technology, Shenzhen, China
[2] Pengcheng Laboratory, China
[3] UQMM Lab, University of Queensland, Australia
zacharymengwk@gmail.com
darrenzz219@gmail.com

## Abstract

Linear attention has emerged as a promising alternative to softmax-based attention, leveraging kernelized feature maps to reduce complexity from quadratic to linear in sequence length. However, the non-negative constraint on feature maps and the relaxed exponential function used in approximation lead to significant information loss compared to the original query-key dot products, resulting in less discriminative attention maps with higher entropy. To address the missing interactions driven by negative values in query-key pairs, we propose a polarity-aware linear attention mechanism that explicitly models both same-signed and opposite-signed query-key interactions, ensuring comprehensive coverage of relational information. Furthermore, to restore the spiky properties of attention maps, we provide a theoretical analysis proving the existence of a class of element-wise functions (with positive first and second derivatives) that can reduce entropy in the attention distribution. For simplicity, and recognizing the distinct contributions of each dimension, we employ a learnable power function for rescaling, allowing strong and weak attention signals to be effectively separated. Extensive experiments demonstrate that the proposed PolaFormer improves performance on various vision tasks, enhancing both expressiveness and efficiency by up to 4.6%. Code is available at https://github.com/ZacharyMeng/PolaFormer.

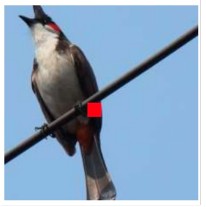 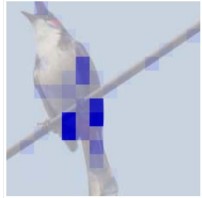 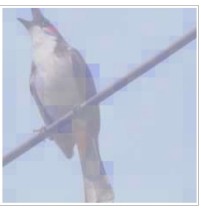 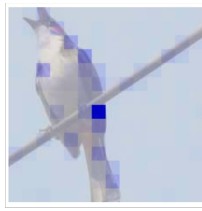 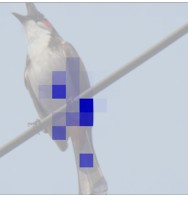

Original Image    Softmax Attention    Linear Attention    Flatten    PolaFormer

Figure 1: **Attention weight visualization.** Unlike prior linear attention approaches *((Katharopoulos et al., 2020) the 3rd and (Han et al., 2023a) 4th plots)* that generate uniform responses, the proposed PolaFormer captures a more accurate query-key interaction with lower entropy, closely resembling softmax while maintaining linear complexity.

## 1 Introduction

Transformers have demonstrated remarkable success across a broad range of vision tasks (Yuan et al., 2021b; Cai et al., 2022). The core component, dot-product attention with softmax normalization, enables transformers to capture long-range dependencies effectively. However, this comes at the cost of quadratic complexity $\mathcal{O}(N^2)$ in relation to the sequence length $N$, resulting in considerable computational overhead particularly when processing long-sequence videos or high-resolution

---

*Correspondence to Zheng Zhang <darrenzz219@gmail.com>

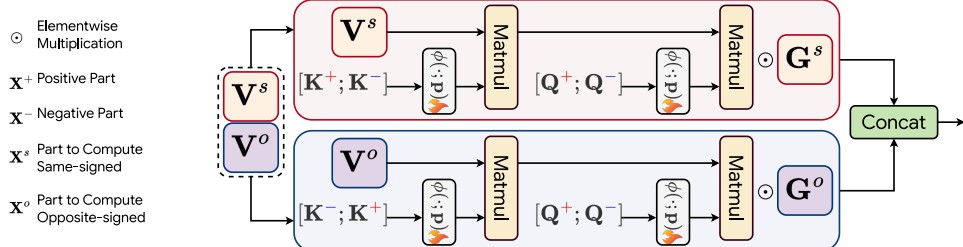

Figure 2: **The overall framework of PolaFormer**. Our framework explicitly separates query-key pairs based on their polarity into two distinct streams, with scaled outputs controlled by the learnable sign-aware matrices $\mathbf{G}^s$ and $\mathbf{G}^o$ for same-signed and opposite-signed components, respectively. A channel-wise power function with the learnable exponent $\mathbf{p}$ is employed to learn the rescaling process, capturing the sharpness characteristic of softmax.

images. This limits their efficiency in resource-constrained environments, making practical deployment difficult in such scenarios.

To mitigate this challenge, various methods have been proposed to *accelerate* attention computation. Techniques such as localized or sparse attention reduce the number of tokens or key-value pairs by restricting attention to smaller windows or sparser patterns, thereby lowering the overall computational costs. While effective, these methods often sacrifice important contextual information, leading to unstable convergence behaviors and performance degradation. As a more principled solution, linear attention (Katharopoulos et al., 2020) replaces the $\mathrm{Softmax}$ operation in the query-key dot product with kernalized feature maps, effectively reducing time and space complexity from $\mathcal{O}(N^2 d)$ to $\mathcal{O}(N d^2)$, where $d$ denotes the feature map dimension. Recent advances in linear attention have centered on designing two key components, *i.e.*, (1) *non-negative feature maps* such as $\mathrm{ELU} +1$ (Katharopoulos et al., 2020) and $\mathrm{ReLU}$ (Qin et al., 2022) and (2) *kernel functions* including Gaussian kernels (Chen et al., 2021), Laplace kernels (Verma, 2021) and polynomial kernels (Kacham et al., 2024), to preserve the core properties of the original $\mathrm{Softmax}$ function while improving computational efficiency.

Despite the efficiency gains, linear attention still falls short in expressive capacity compared to softmax-based attention: As illustrated in Figure 1, it often yields more *uniform* attention weights across query-key pairs, thus resulting in reduced specificity. For instance, when querying a particular region like *bird wing*, linear attention tends to activate key tokens from unrelated areas (*e.g.*, *poles*) equally, introducing noise that disrupts downstream vision tasks. Our analysis identifies two primary causes for this shortfall, both stemming from **information loss** during the $\mathrm{Softmax}$ approximation:

(1) **Loss of Negative Values.** Linear attention models that rely on non-negative feature maps, such as $\mathrm{ReLU}$, fail to maintain consistency with the original query-key dot product. These feature maps retain only *positive-positive* interactions, while crucial *negative-negative* and *positive-negative* interactions are completely dropped. This selective representation limits the model's ability to capture a comprehensive range of relationships, leading to diminished expressiveness and reduced discriminative power in the resulting attention maps.

(2) **Loss of Attention Spikeness.** Without the exponential scaling of softmax, linear attention leads to more uniform weight distributions and lower entropy. This uniformity weakens the model's ability to distinguish between strong and weak query-key pairs, impairing its focus on important features and reducing performance in tasks requiring fine detail.

In this work, we propose a polarity-aware linear attention (**PolaFormer**) mechanism, designed to address the limitations of prior linear attention models by incorporating the previously omitted negative interactions. Unlike traditional approaches that only preserve positive-positive query-key interactions, PolaFormer explicitly separates query-key pairs based on their polarity—handling a full spectrum of same-signed (*positive-positive*, *negative-negative*) and opposite-signed (*positive-negative*, *negative-positive*) interactions as shown in Figure 2. These interactions are processed in two streams, allowing for a more accurate reconstruction of the original softmax attention weights. To avoid unnecessary complexity, we split the value vector along the channel dimension, handling both types of interactions without introducing additional learnable parameters. The outputs are then

concatenated and scaled with a learnable sign-aware matrix, ensuring a faithful reconstruction of query-key relationships.

To mitigate the issue of uniform attention weights commonly observed in linearized attention, we provide a theoretical foundation showing that an element-wise function can rescale the query-key responses to reduce entropy, provided the function has positive first and second derivatives. This insight helps clarify why previous feature maps such as $\mathrm{ReLU}$ and $\mathrm{ELU}$ tend to elevate entropy, leading to overly smoothed attention distributions. For simplicity, we employ a channel-wise learnable power function for rescaling, which retains the sharpness of the exponential function inherent in $\mathrm{Softmax}$. This enables the model to capture spiky attention peaks, improving its ability to distinguish between strong and weak responses. Together, these enhancements offer a more robust solution to bridging the gap between linearized and softmax-based attention. We conduct extensive experiments on various vision tasks and the Long Range Arena benchmark (Tay et al., 2021), demonstrating that our model enhances performance by up to 4.6% while preserving a superior balance between expressive capability and efficiency.

## 2 RELATED WORK

**Efficient Vision Transformers.** By cutting images into smaller patches and processing them as a sequence, Vision Transformers (ViT) (Dosovitskiy et al., 2021) successfully transfer transformer models (Vaswani et al., 2017) from language tasks to vision tasks, and have achieved remarkable results. However, the quadratic complexity of the self-attention mechanism in ViT makes it expensive to train. Existing works have made various improvements to ViT for computational efficiency. Swin-Transformer (Liu et al., 2021) introduces a shifted windows scheme to limit self-attention computation to local windows. Pyramid Vision Transformer (PVT) (Wang et al., 2021) uses a progressive shrinking pyramid to reduce the computations of feature maps. Deit (Touvron et al., 2021) enables models to achieve competitive performance without pretraining on large datasets by utilizing designed tokenization mechanisms and training strategies. However, these improvements do not solve the bottleneck of the self-attention mechanism, and quadratic complexity, thus the training cost is still unaffordable as the model scale increases. To address this issue, VMamba (Liu et al., 2024) extracts the information of the picture based on the spatial state model (SSM) encoding through serializing and scanning the picture, at the same time it inherits the linear complexity of SSM. VHeat (Wang et al., 2024) conceptualize image patches as heat sources and simulate the conduction process, and utilize DCT and IDCT operations to reduce the complexity to $\mathcal{O}(N^{1.5})$. These methods have just been proposed and are not yet as widely validated and deployed at scale as Transformers,their model performance is also not significantly higher than the other models.

**Linear Attention.** Sub-quadratic transformers focus on alleviating the inefficiency of the standard self-attention mechanism due to the softmax function and its quadratic complexity. A preferable solution is to use kernel-based similarities to reduce the complexity by approximating the softmax operator. The initial linear attention (Katharopoulos et al., 2020) proposes to substitute the $\mathrm{Softmax}$ function with a linear dot-product of kernel feature maps, which facilitates reducing the complexity from $\mathcal{O}(N^2)$ to $\mathcal{O}(N)$. Following this $\mathrm{Softmax}$-*free* scheme, some variants of linear attention have been proposed by employing different kernel functions, such as $\mathrm{ReLU}(\cdot)$ (Shen et al., 2021) and $1 + \mathrm{ELU}(\cdot)$ (Katharopoulos et al., 2020). Moreover, to fulfill the non-negative and distribution properties of attention matrix, Cosformer (Qin et al., 2022) combines the ReLU function and cos-based re-weighting mechanism to enhance the self-attention weighs with locality inductive biases. FLatten Transformer (Han et al., 2023a) extends $\mathrm{ReLU}(\cdot)$ with power operation to maintain both properties of attention weights, *i.e.,* non-negative and low-entropy. It is a practical way to use power function to calculate the inner product to approximate exp, which is similar to the use of power function to approximate max-pooling proposed in R-MAC (Tolias et al., 2016). Recently, Agent Attention (Han et al., 2023b), a claimed generalized linear attention, introduces $n$ agent tokens to aggregate features based on a combination of Softmax and linear attention with $\mathcal{O}(Nnd)$ complexity. As both $N$ and $n$ increase simultaneously with the model size, the complexity of the generalized linear attention is not absolutely linear with respect to $N$. Notably, the balanced performance still relies on the softmax operator and additional agent tokens, which violates the original premise of linear attention, *i.e.,* softmax-free and linear complexity. Current kernel functions either suffer from performance degradation or introduce excessive computational overhead. We observed significant information loss in comparison to original query-key dot products due to the non-negative constraint

on attention weights and the intricate kernel designs aimed at achieving low entropy. This issue will be further addressed in the following sections of this work.

## 3 PRELIMINARY

In this section, we first highlight the inefficiency of the standard self-attention mechanism, followed by a discussion of the variants of existing linear attention methods.

### 3.1 LOW EFFICIENCY OF SELF-ATTENTION MECHANISM

Consider a sequence $\mathbf{x} \in \mathbb{R}^{N \times D}$ of token length $N$ and dimension $D$. $\mathbf{x}$ is devided into $h$ heads, the dimension of each head is $d$. In each head, tokens at various positions are collectively attended to capture long-range dependencies. The output $\mathbf{O} = \{\mathbf{o}_t\}_{t=1}^{N} \in \mathbb{R}^{N \times d}$ can be formulated as:

$$\mathbf{O} = \text{Softmax}(\frac{\mathbf{Q}\mathbf{K}^\top}{\sqrt{d}})\mathbf{V}, \ \mathbf{o}_t = \frac{\sum_{i=1}^{N} \exp(\mathbf{q}_t \mathbf{k}_i^\top / \sqrt{d})}{\sum_{j=1}^{N} \exp(\mathbf{q}_t \mathbf{k}_j^\top / \sqrt{d})} \mathbf{v}_i. \tag{1}$$

Here, the query, key, and value vectors of dimension $d$ are obtained by linearly projecting the inputs with three learnable matrices $\mathbf{Q} = \{\mathbf{q}_t\}_{t=1}^{N}$, $\mathbf{K} = \{\mathbf{k}_t\}_{t=1}^{N}$, $\mathbf{V} = \{\mathbf{v}_t\}_{t=1}^{N} \in \mathbb{R}^d$. For each head, the complexity of self-attention is $\mathcal{O}(N^2 d)$, making the mechanism inefficient for long sequences.

### 3.2 KERNEL-BASED LINEAR ATTENTION

To mitigate the efficiency bottlenecks of standard self-attention, kernel-based linear attention mechanisms (Katharopoulos et al., 2020) have been proposed, which decompose the similarity function into dot products of feature maps. Following the notations in (Choromanski et al., 2021; Chen et al., 2021), we define $\mathbf{SM}(\mathbf{q}, \mathbf{k}) = \exp(\mathbf{q}_i \mathbf{k}_j^\top)$ as the softmax kernel function. Mathematically, linear attention aims to use $\phi(\mathbf{q}_i)\phi(\mathbf{k}_j)^\top$ to approximate $\mathbf{SM}(\cdot, \cdot)$, where the feature map $\phi(\cdot) : \mathbb{R}^d \mapsto \mathbb{R}^{d'}$ is applied row-wise to the query and key matrices. As a result, the $t$-th row of attention output $\mathbf{o}_t$ can be rewritten as,

$$\mathbf{o}_t = \frac{\sum_{i=1}^{N} \phi(\mathbf{q}_t)\phi(\mathbf{k}_i)^\top \mathbf{v}_i}{\sum_{j=1}^{N} \phi(\mathbf{q}_t)\phi(\mathbf{k}_j)^\top} = \frac{\phi(\mathbf{q}_t) \sum_{i=1}^{N} \phi(\mathbf{k}_i)^\top \mathbf{v}_i}{\phi(\mathbf{q}_t) \sum_{j=1}^{N} \phi(\mathbf{k}_j)^\top}. \tag{2}$$

By leveraging the associative property of matrix multiplication, the complexity per head is reduced to $\mathcal{O}(Nd'^2)$, which scales linearly with the sequence length.

**Choices of Feature Map** $\phi(\cdot)$. The primary distinction between various linear attention methods lies in the choice of feature maps $\phi(\cdot)$. Considering $\mathbf{SM}(\cdot, \cdot)$ is a PSD kernel function and the chosen feature map $\phi$ must satisfy two properties:

1. **Non-negativity.** To preserve the non-negative values in the approximation of $\mathbf{SM}$, previous methods utilize activation functions like $\phi(\mathbf{x}) = 1 + \text{ELU}(\mathbf{x})$ (Katharopoulos et al., 2020) or $\phi(\mathbf{x}) = \text{ReLU}(\mathbf{x})$ (Qin et al., 2022; Han et al., 2023a). Other approaches connect $\mathbf{SM}$ with Gaussian kernel that uses $\phi(x) = \exp(\frac{\|\mathbf{x}^2\|}{2})$, incorporating trigonometric or random positive features.

2. **Low Entropy.** It has been observed the attention-weights distribution in standard Transformers tends to be more "spiky" in linear ones, exhibiting lower entropy (Zhang et al., 2024a). To rescale the query-key dot products back to the original magnitudes, techniques such as Taylor expansion (Keles et al., 2023) or higher norms on the numerical value of query and key (Han et al., 2023a) have been employed.

However, using non-negative feature maps inherently results in the loss of information from the original *negative* values, which may carry important information in the original dot product calculation. This leads to discontinuities in the linear attention map compared to the standard attention. Furthermore, existing rescaling strategies (Han et al., 2023a) manually select a fixed norm across all dimensions, *i.e.*, $\phi(\mathbf{x}) = f_p(\text{ReLU}(\mathbf{x}))$, where $f_p(\mathbf{x}) = \frac{\|\mathbf{x}\|}{\|\mathbf{x}^p\|}\mathbf{x}^p$. This fixed norm $p$ may not be optimal across different datasets.

## 4 PROPOSED APPROACH

In this section, we present a novel polarity-aware attention mechanism that accurately captures query-key interactions without incurring additional computational overhead. Our method incorpo-

rates a learnable dimension-wise power function that dynamically rescales the magnitudes of same- and opposite-signed components, effectively reducing entropy in the linear attention.

## 4.1 POLARITY-AWARE ATTENTION

The key idea behind polarity-aware attention is to address the limitations of existing linear attention mechanisms, which often discard valuable information from negative components. We start by decomposing the query vector $\mathbf{q} = \{q_i\}_{i \in [d]} \in \mathbb{R}^d$ and key vector $\mathbf{k} = \{k_i\}_{i \in [d]} \in \mathbb{R}^d$ element-wise into their positive and negative components:

$$\mathbf{q} = \mathbf{q}^+ - \mathbf{q}^-, \quad \mathbf{k} = \mathbf{k}^+ - \mathbf{k}^-, \tag{3}$$

where $\mathbf{q}_i^+ = \max(q_i, 0)$ and $\mathbf{q}_i^- = \max(-q_i, 0)$, representing the positive and negative parts of $\mathbf{q}$, respectively, and similarly for $\mathbf{k}$. Substituting these decompositions into the inner product of $\mathbf{q}$ and $\mathbf{k}$ gives:

$$\langle \mathbf{q}, \mathbf{k} \rangle = \langle \mathbf{q}^+, \mathbf{k}^+ \rangle + \underbrace{\langle \mathbf{q}^-, \mathbf{k}^- \rangle - \langle \mathbf{q}^+, \mathbf{k}^- \rangle - \langle \mathbf{q}^-, \mathbf{k}^+ \rangle}_{\texttt{neglected negatives}} \tag{4}$$

The first two terms capture the similarity between *same-signed* components, while the latter two terms represent interactions between *opposite-signed* components. Previous linear attention approaches, such as ReLU-based feature maps, eliminate negative components by mapping them to zero, resulting in significant information loss when approximating query-key dot products.

To address this, our polarity-aware attention mechanism separates query-key pairs based on their polarity, computing their interactions *independently*. The attention weights are calculated as follows:

$$\begin{aligned}
\mathbf{SM}(\mathbf{q}, \mathbf{k}^\top) &= \exp(\mathbf{q}\mathbf{k}^\top) \\
&\approx \left( \phi(\mathbf{q}^+)\phi(\mathbf{k}^+)^\top + \phi(\mathbf{q}^-)\phi(\mathbf{k}^-)^\top \right) - \left( \phi(\mathbf{q}^+)\phi(\mathbf{k}^-)^\top + \phi(\mathbf{q}^-)\phi(\mathbf{k}^+)^\top \right).
\end{aligned} \tag{5}$$

This formulation recovers the information embedded in both positive and negative components.

**Learnable Polarity-aware Mixing.** While this formulation captures key information carried by both same-signed and opposite-signed components, directly subtracting opposite-signed similarities can violate non-negativity constraints, leading to unstable training and suboptimal performance. To avoid the pitfalls of subtractive operation, we instead resort to a learnable mixing mechanism that weighs the contributions of same-signed and opposite-signed query-key similarities.

More concretely, we split each value vector $\mathbf{v} \in \mathbb{R}^{N \times d}$ along the $d$ dimension into two halves to separately handle same- and opposite-signed response, *i.e.*, $\mathbf{v} = [\mathbf{v}^s; \mathbf{v}^o]$, where both $\mathbf{v}^s$ and $\mathbf{v}^o$ have a dimensionality of $d/2$. The output attention is then computed as:

$$\mathbf{o}_t = \left[ \frac{\phi([\mathbf{q}_t^+; \mathbf{q}_t^-]) \sum_{i=1}^N \phi([\mathbf{k}_i^+; \mathbf{k}_i^-])^\top \mathbf{v}_i^s}{\phi([\mathbf{q}_t^+; \mathbf{q}_t^-]) \sum_{j=1}^N \phi([\mathbf{k}_j^+; \mathbf{k}_j^-])^\top} \odot \mathbf{G}^s; \frac{\phi([\mathbf{q}_t^+; \mathbf{q}_t^-]) \sum_{i=1}^N \phi([\mathbf{k}_i^-; \mathbf{k}_i^+])^\top \mathbf{v}_i^o}{\phi([\mathbf{q}_t^+; \mathbf{q}_t^-]) \sum_{j=1}^N \phi([\mathbf{k}_j^-; \mathbf{k}_j^+])^\top} \odot \mathbf{G}^o \right], \tag{6}$$

(a) Pearson Correlation  (b) Weight Distribution

Figure 3: Visualizations of weights in $\mathbf{G}^s$ and $\mathbf{G}^o$.

where $[\cdot, \cdot]$ denotes concatenation operation. $\mathbf{G}^s \in \mathbb{R}^{N \times \frac{d}{2}}$ and $\mathbf{G}^o \in \mathbb{R}^{N \times \frac{d}{2}}$ are two learnable polarity-aware coefficients matrices applied with element-wise multiplication, which are expected to learn the complementary relationship between same-signed and opposite-signed values. As shown in Figure 3, there is a clear negative correlation and value discrepancy between the weights learned in $\mathbf{G}^s$ and $\mathbf{G}^o$, which evidences our learnable mixing strategy compensates for the relaxed subtraction operation in Equation (5).

**Low-Rank SM.** Previous theoretical work (Verma, 2021) has shown that $\mathbf{SM}$ is inherently low-rank, particularly in higher layers where the spectrum distribution becomes more skewed. This property can lead to degenerate solutions when learning value vectors, especially when compact representations are required to accommodate polarity-aware information in our case. We explore various techniques such as depthwise and deformable convolutions to increase the rank, which can refer to the ablation study in Section 5.4.

### 4.2 Reducing Entropy in Linear Attention via Learnable Power Functions

Softmax-free linear attention mechanisms often exhibit higher entropy compared to softmax-based attention, leading to less sharp value vector attention, which is detrimental to tasks requiring precise attention. To recover the low entropy characteristics observed in softmax-based attention, we reinterpret each row in $\mathbf{SM}(\mathbf{q}, \mathbf{k}^\top)$ as a generalized unnormalized positive sequence $\mathbf{x} = (x_1, ..., x_N)$ and analyze its entropy using our proposed positive sequence entropy (PSE) measure, defined as:

**Definition 1** (Positive Sequence Entropy (PSE)). *Let a sequence $\mathbf{x} = (x_1, ..., x_N)$, in which $x_i \geq 0$, $i = 1, \ldots, N$, and $s = \sum_{i=1}^N x_i > 0$. Then the entropy of this positive sequence is defined by:*

$$\text{PSE}(\mathbf{x}) = -\sum_{i=1}^N \frac{x_i}{s} \log(\frac{x_i}{s}),\, s = \sum_{i=1}^N x_i. \tag{7}$$

With $\text{PSE}(\cdot)$ defined, we now seek a function $g(\cdot)$ that can be applied element-wise to $\phi(\mathbf{q}^i)$ and $\phi(\mathbf{K}) = [\phi(\mathbf{k}^1), \ldots, \phi(\mathbf{k}^N)]$ such that the PSE of the $i$-th row of the linear attention map is reduced. The following theorem formalizes the *conditions* under which this reduction in PSE can be achieved.

**Theorem 1.** *Let $\mathbf{x}, \mathbf{y}^n \in \mathbb{R}^d$ for $n = 1, \ldots N$, and let $g : [0, +\infty) \mapsto [0, +\infty)$ be a differentiable function satisfying the condition $g'(x) > 0$ and $g''(x) > 0$ for all $x > 0$. Then, there exists such a function $g$ such that the PSE of the transformed sequence is strictly less than that of the original sequence. Specifically, we have:*

$$\text{PSE}(\langle g(\mathbf{x}), g(\mathbf{y}^1)\rangle, \ldots, \langle g(\mathbf{x}), g(\mathbf{y}^N)\rangle) < \text{PSE}(\langle \mathbf{x}, \mathbf{y}^1 \rangle, \ldots, \langle \mathbf{x}, \mathbf{y}^N \rangle). \tag{8}$$

*Proof and supporting lemmas are provided in Section A.1.*

This theorem also provides insights into why commonly used feature maps $\phi$ such as ReLU or ELU $+1$ fail to reduce entropy effectively, as they do not satisfy the necessary conditions of having both a positive first and second derivative across their entire domain.

To select a suitable function $g$, There exists a wide variety of functions $g$ that meet these conditions. However, for the sake of model simplicity and efficiency, we opt for the most straightforward choice: a power function with an exponent greater than 1. Additionally, as different dimensions may contribute *unequally* to the similarity computation, we design learnable exponents to capture the varying importance of each dimension, formalized as follows:

$$\mathbf{p} = 1 + \alpha \,\text{sigmoid}(w_1, \ldots, w_d),\, g(\mathbf{x}; \mathbf{p}) = (x_1^{p_1}, \ldots, x_d^{p_d}) \tag{9}$$

where $\alpha > 0$ is a hyper-parameter scaling factor and $[w_1, \ldots, w_d]$ are learnable parameters. Therefore, the feature map in our linear attention can be expressed as $\phi(\mathbf{x}^+) = g(\text{ReLU}(\mathbf{x}); \mathbf{p})$ and $\phi(\mathbf{x}^-) = g(\text{ReLU}(-\mathbf{x}); \mathbf{p})$, where $\mathbf{x}$ refers to either $\mathbf{q}$ or $\mathbf{k}$.

**Complexity Analysis.** We now analyze the complexity complexity of PolaFormer and demonstrate its linear complexity. Let $d$ denote the number of channels, $d'$ the dimensionality after kernelized, and $k$ the kernel size of convolution ($d' = d$ since $g()$ does just a element-wise mapping). The computational cost for query, key, value, coefficients $\mathbf{G}^s$ and $\mathbf{G}^o$ and outputs projections is $5Nd^2$. Performing matrix multiplication for $(\mathbf{Q}, \mathbf{K}, \mathbf{V})$ across each head requires $4Ndd'$. The convolution operation contributes $k^2Nd$, while the element-wise multiplication of polarity-aware coefficients $\mathbf{G}^s$ and $\mathbf{G}^o$ requires $Nd$ computations. Summarizing these components, the total complexity of PolaFormer is given in Equation (10), confirming its linear complexity *w.r.t.* sequence length $N$.

$$\Omega = \underbrace{5Nd^2}_{\text{Proj}} + \underbrace{4Ndd'}_{\text{Pola Attn}} + \underbrace{k^2Nd}_{\text{Conv}} + \underbrace{Nd}_{\text{coeff}} \tag{10}$$

## 5 Experiments

In this section, we evaluate our PolaFormer model on three tasks: image classification on ImageNet-1K (Deng et al., 2009), object detection and instance segmentation on COCO (Lin et al., 2014), and semantic segmentation on ADE20K (Zhou et al., 2019), comparing its performance with previous efficient vision models. Additionally, we assess PolaFormer on the Long Range Arena (LRA) task (Tay et al., 2021) to compare against other linear attention models. We first train PolaFormer from scratch on the image classification task, then fine-tune the pre-trained model on ADE20K dataset for segmentation and COCO dataset for detection. The models were pretrained on 8 NVIDIA A800 GPUs and fine-tuned on 8 NVIDIA RTX A6000 and 8 NVIDIA RTX 3090 GPUs.

Table 1: Comparison of various linear attention methods relative to the original models (DeiT-T and Swin-T) on the ImageNet-1K dataset, with the best results highlighted in boldface.

| METHOD | RESO | PARAMS | FLOPs | ACC(%) |
|---|---|---|---|---|
| DeiT (Touvron et al., 2021) | $224^2$ | 5.7M | 1.1G | 72.2 |
| DeiT-EfficientAttn (Shen et al., 2021) | $224^2$ | 5.7M | 1.1G | 70.2 |
| DeiT-HydraAttn (Bolya et al., 2022) | $224^2$ | 5.7M | 1.1G | 68.3 |
| DeiT-EnhancedAttn (Cai et al., 2022) | $224^2$ | 5.8M | 1.1G | 72.9 |
| DeiT-AngularAttn (You et al., 2023) | $224^2$ | 5.7M | 1.1G | 70.8 |
| DeiT-FLattenAttn (Han et al., 2023a) | $224^2$ | 6.1M | 1.1G | 74.1 |
| DeiT-MobiAttn (Yao et al., 2024) | $224^2$ | 5.7M | 1.2G | 73.3 |
| DeiT-PolaFormer | $224^2$ | 6.1M | 1.2G | $\mathbf{74.6}_{+2.4}$ |
| Swin (Liu et al., 2021) | $224^2$ | 28M | 4.4G | 81.2 |
| Swin-HydraAttn (Bolya et al., 2022) | $224^2$ | 29M | 4.5G | 80.7 |
| Swin-EfficientAttn (Shen et al., 2021) | $224^2$ | 29M | 4.5G | 81.0 |
| Swin-LinearAngularAttn (You et al., 2023) | $224^2$ | 29M | 4.5G | 79.4 |
| Swin-EnhancedAttn (Cai et al., 2022) | $224^2$ | 29M | 4.5G | 81.8 |
| Swin-FLattenAttn (Han et al., 2023a) | $224^2$ | 29M | 4.5G | 82.1 |
| Swin-PolaFormer | $224^2$ | 29M | 4.5G | $\mathbf{82.6}_{+1.4}$ |

Figure 4: Efficiency analysis with Accuracy vs. FLOPs and Accuracy vs. Runtime curves on the ImageNet-1K dataset.

## 5.1 IMAGENET-1K CLASSIFICATION

The ImageNet-1K (Deng et al., 2009) dataset is the widely used dataset for image classification tasks, containing 1,000 categories and over 1.2 million training images. We comprehensively assess our model's performance using Top-1 accuracy, and compare it against recent state-of-the-art efficient Vision Transformer (ViT) models. Specifically, we selected four representative ViT backbones: DeiT (Touvron et al., 2021), PVT (Wang et al., 2021), PVTv2 (Wang et al., 2022) and Swin-Transformer (Liu et al., 2021). We replaced their self-attention modules with the our proposed polarity-aware attention module and trained these Pola-variants from scratch on ImageNet-1K.

**Results.** The experimental results are presented in Table 1 and Table 5, consistently showing that our model outperforms the baseline models. For instance, in Table 1, our DeiT-T-PolaFormer surpasses other DeiT variants from 0.5% to 6.3%. In Table 5, the PVT-T/S-PolaFormer obtain an increase of 3.7% and 2.1% comparing with the corresponding baseline with comparable FLOPs. Additionally, our method integrated in Swin and PVTv2 achieves a better balance between performance and efficiency. These results demonstrate that the PolaFormer enhances the expressive capability of the attention mechanism and can be widely applied in various attention-based models.

**Efficiency Analysis.** We visualize the efficiency comparison between the proposed PolaFormer and other linear attention approaches with similar FLOPs in the first two plots of Figure 4. The results show that our model can achieve comparable performance with significantly less computation. Furthermore, we evaluate the inference speed of PolaFormer. To be specific, we test the PVT-PolaFormer and Swin-PolaFormer on RTX3090 and RTXA6000 platforms, as shown in the third and forth plots of Figure 4. PVT-PolaFormer achieves $1.15\times$ and $1.12\times$ faster inference speed and Swin-PolaFormer achieves $1.32\times$ and $1.29\times$ faster, both with comparable or higher accuracy. These figures highlight the excellent trade-off between accuracy and latency that our model provide.

## 5.2 OBJECT DETECTION AND INSTANCE SEGMENTATION

We further validate the effectiveness of the proposed approach across various vision tasks, including object detection task on the COCO dataset (Lin et al., 2014), which contains over 118K training images and 5K validation images. We integrate Pola-Swin and Pola-PVT separately as the backbone into Mask-RCNN (M) (He et al., 2017), RetinaNet (R) (Lin et al., 2017) and Cascade Mask

Table 2: Object detection and instance segmentation results on the COCO dataset. The "Type" column specifies the detector used: R represents RetinaNet, M for Mask R-CNN, and C for Cascade Mask R-CNN. For semantic segmentation on the ADE20K dataset (last column), two encoder types are employed: S corresponds to Semantic FPN, and U refers to UperNet.

| METHOD | SCH. | TYPE | DETECTION AND INSTANCE SEGMENTATION | | | | | | SEMANTIC SEG | |
|---|---|---|---|---|---|---|---|---|---|---|
| | | | $AP^b$ | $AP^b_{50}$ | $AP^b_{75}$ | $AP^m$ | $AP^m_{50}$ | $AP^m_{75}$ | TYPE | mIoU(%) |
| PVT-T | 1× | R | 36.7 | - | - | - | - | - | S | 35.7 |
| | 1× | M | 36.7 | 59.2 | 39.3 | 35.1 | 56.7 | 37.3 | | |
| PVT-T-Flatten | 1× | R | - | - | - | - | - | - | S | 37.2 |
| | 1× | M | 38.2 | 61.6 | 41.9 | 37.0 | 57.6 | 39.0 | | |
| PVT-T-PolaFormer | 1× | R | $39.5_{+2.8}$ | **60.1** | **42.1** | - | - | - | S | $38.3_{+2.6}$ |
| | 1× | M | $40.4_{+3.7}$ | $62.4_{+3.2}$ | $43.9_{+4.6}$ | $37.4_{+2.3}$ | $59.4_{+2.7}$ | $40.3_{+3.0}$ | | |
| PVT-S | 1× | R | 40.4 | - | - | - | - | - | S | 39.8 |
| | 1× | M | 40.4 | 62.9 | 43.8 | 37.8 | 60.1 | 40.3 | | |
| PVT-S-PolaFormer | 1× | R | $43.2_{+2.8}$ | **64.1** | **46.4** | - | - | - | S | $41.0_{+1.2}$ |
| | 1× | M | $43.9_{+3.5}$ | $66.1_{+3.2}$ | $47.9_{+4.1}$ | $40.2_{+2.4}$ | $63.1_{+3.0}$ | $43.0_{+2.7}$ | | |
| Swin-T | 1× | M | 43.7 | 66.6 | 47.7 | 39.8 | 63.3 | 42.7 | U | 44.5 |
| | 3× | M | 46.0 | 68.1 | 50.3 | 41.6 | 65.1 | 44.9 | | |
| | 3× | C | 50.4 | 69.2 | 54.7 | 43.7 | 66.6 | 47.3 | | |
| Swin-T-FLatten | 1× | M | 44.2 | 67.3 | 48.5 | 40.2 | 63.8 | 43.0 | U | 44.8 |
| | 3× | M | 46.5 | 68.5 | 50.8 | 42.1 | 65.4 | 45.1 | | |
| | 3× | C | 50.8 | 69.6 | 55.1 | 44.1 | 67.0 | 48.1 | | |
| Swin-T-PolaFormer | 1× | M | $44.8_{+1.1}$ | $67.6_{+1.0}$ | $49.1_{+1.4}$ | $40.5_{+0.7}$ | $64.1_{+0.8}$ | $43.5_{+0.7}$ | U | $45.8_{+1.3}$ |
| | 3× | M | $47.0_{+1.0}$ | $68.9_{+0.8}$ | $51.5_{+1.2}$ | $42.3_{+0.7}$ | $66.0_{+0.9}$ | $45.8_{+0.9}$ | | |
| | 3× | C | $51.1_{+0.7}$ | $70.0_{+0.8}$ | $55.6_{+0.9}$ | $44.4_{+0.7}$ | $67.3_{+0.7}$ | $48.3_{+1.0}$ | | |

R-CNN (C) (Cai & Vasconcelos, 2021) implementations and evaluate their performance based on the ImageNet-1k pretrained weights. As shown in Table 2 (left), our model consistently outperforms the original backbones under all settings, achieving notable improvements in all metrics. For instance, our PVT-T-PolaFormer tested with both R and M detectors, surpasses the baselines from 2.3% to 4.6%. Additionally, our Swin-T-PolaFormer achieves 49.1% in $AP^b_{75}$, showing a 1.4% improvement compared to the original Swin-T with M detector. We additionally evaluate PVT-S-PolaFormer with R and M detectors, and Swin-T-PolaFormer with M and C detectors using 1× and 3× schedule. Compared to classification tasks, our model delivers more substantial performance gains on detection, which demands fine-grained attention maps for accurate localization of bounding boxes. Our model captures previously omitted interactions involving negative values and better restores attention maps with appropriate scales, effectively distinguishing between similar and dissimilar query-key relationships.

## 5.3 SEMANTIC SEGMENTATION

A similar phenomenon was observed when fine-tuning our pre-trained model for pixel-wise semantic segmentation tasks on the ADE20K dataset. ADE20K (Zhou et al., 2019) provides a diverse set of annotations for scenes, objects, and object parts, containing 25,000 images of complex scenes with various objects in natural spatial environments. We integrate Pola-Swin and Pola-PVT with ImageNet-1K pre-trained weights into two segmentation models, SemanticFPN (Kirillov et al., 2019) and UperNet (Xiao et al., 2018), using mIoU as the evaluation metric. The results, shown in Table 2 (right), demonstrate a performance improvement in mIoU ranging from 1.2% to 2.6%. These findings further highlight the versatility of our model, showing that it can be effectively fine-tuned and adapted to a wide range of vision tasks.

## 5.4 ABLATION STUDY

Table 3: Ablation study on each module using DeiT-T on ImageNet-1K.

| POLARITY | $\mathbf{G}^s,\mathbf{G}^o$ | DWC | DCN | ACC. (%) |
|---|---|---|---|---|
| ✓ | ✓ | | ✓ | $61.9_{-12.7}$ |
| ✓ | | | | $68.1_{-6.5}$ |
| ✓ | | ✓ | | $72.8_{-1.8}$ |
| ✓ | ✓ | ✓ | | 74.6 |

**Impact of Components.** We evaluate the effectiveness of each component in PolaFormer. As shown in Table 3, to address the low-rank issue of the attention map, we examine the impact of incorporating deformable convolutions (DCN) and depth-wise convolutions (DWC) in row 1 and row 4, respectively. DWC demonstrates better adaptability, achieving an accuracy of 74.6%. It is important to note that our model is agnostic to the choice of convolution modules. Furthermore, adopting polarity coefficients $\mathbf{G}^s$ and $\mathbf{G}^o$ yields a 1.8% improvement in row 3 and 4, indicating that the model effectively learns the complementary relationship between same-signed and opposite-signed values.

Table 4: Comparisons (%) between the proposed PolaFormer and other linear attention models on LRA, with the best results are highlighted in boldface.

| MODEL | TEXT | LISTOPS | RETRIEVAL | PATHFINDER | IMAGE | AVERAGE |
|---|---|---|---|---|---|---|
| Transformer | 61.55 | 38.71 | 80.93 | 70.39 | 39.14 | 58.14 |
| LocalAttn | 52.98 | 15.82 | 53.39 | 66.63 | 41.46 | 46.06 |
| LinearTrans. | 65.90 | 16.13 | 53.09 | 75.30 | 42.34 | 50.55 |
| Reformer | 56.10 | 37.27 | 53.40 | 68.50 | 38.07 | 50.67 |
| Performer | 65.40 | 18.01 | 53.82 | **77.05** | **42.77** | 51.41 |
| Synthesizer | 61.68 | 36.99 | 54.67 | 69.45 | 41.61 | 52.88 |
| Longformer | 62.85 | 35.63 | 56.89 | 69.71 | 42.22 | 53.46 |
| Informer | 62.13 | 37.05 | 79.35 | 56.44 | 37.86 | 54.57 |
| Bigbird | 64.02 | 36.05 | 59.29 | 74.87 | 40.83 | 55.01 |
| Linformer | 57.29 | 36.44 | 77.85 | 65.39 | 38.43 | 55.08 |
| Kernelized | 60.02 | 38.46 | 82.11 | 69.86 | 32.63 | 56.62 |
| Cosformer | 63.54 | 37.2 | 80.28 | 70.00 | 35.84 | 57.37 |
| Nystrom | 62.36 | 37.95 | 80.89 | 69.34 | 38.94 | 57.90 |
| Skyformer | 64.70 | 38.69 | 82.06 | 70.73 | 40.77 | 59.39 |
| Hedgehog | 64.60 | 37.15 | **82.24** | 74.16 | 40.15 | 59.66 |
| PolaFormer$_{\alpha=3}$ | **73.06** | 37.35 | 80.50 | 70.53 | 42.15 | **60.72** |
| PolaFormer$_{\alpha=5}$ | 72.33 | **38.76** | 80.37 | 68.98 | 41.91 | 60.47 |
| PolaFormer$_{\alpha=7}$ | 71.93 | 37.60 | 81.47 | 69.09 | **42.77** | 60.57 |

**Comparison with Other Linear Attention.** To compare with other linear attention models, we evaluate our PolaFormer on the Long Range Arena (LRA) (Tay et al., 2021) task, which is composed with five tasks: ListOps (Nangia & Bowman, 2018), Text Classification on IMDb review dataset (Maas et al., 2011), Document Retrieval on AAN dataset (Radev et al., 2013), Pathfinder (Linsley et al., 2018), and Image Classification on CIFAR-10 (Krizhevsky, 2009). The sequence length in both TEXT and RETRIEVAL tasks is 4k, in LISTOPS is 2k and in PATHFINDER and IMAGE is 1k. Following the setup of Skyformer (Chen et al., 2021), we adopt a comparable number of parameters and train the entire model end-to-end with the task-specific losses. Results for different scaling factors $\alpha$ of PolaFormer are shown in the bottom rows of Table 4. PolaFormer$_{\alpha=3}$ obtains the highest accuracy 73.06% in Text Classification task, PolaFormer$_{\alpha=5}$ has achieved the state-of-the-art results in ListOps task for 38.76% accuracy and the PolaFormer$_{\alpha=7}$ gains the best performance in Image Classification task with an accuracy of 42.77%. It is worth mentioning that PolaFormer$_{\alpha=3}$ achieves the highest overall scores on LRA benchmark, with all variants outperforming other linear attention models. Our model achieves better scores in linear complexity relying on its extraction ability and higher rank, showing great potential in both NLP and CV tasks.

**Impact of Learnable Scaling.** In Equation (9), we introduce the scaling factor $\alpha$ in our learnable power function, and analyze the effect of exponent **p** on the model performance. The value of $\alpha$ primarily depends on the model size and context length. As the model size increases, a larger $\alpha$ is required to effectively select the most relevant tokens from long sequences, thereby reducing information entropy. We evaluate the model with $\alpha = 3, 5, 7$ on the LRA task, shown at the bottom of Table 4. Although PolaFormer$_{\alpha=3}$ achieves the best performance, in practice, for classification tasks, the results are relatively insensitive to variations in $\alpha$, with a difference of no more than 2%.

## 6 CONCLUSION

In this work, we presented PolaFormer, a novel efficient transformer with linear complexity. Our PolaFormer is built on two properties of the original softmax attention: (i) making each element of the attention weight non-negative and (ii) making attention weight spikier. To fulfill these properties, we computed the similarity in a polarity-aware form to avoid neglecting negatives; theoretically, we proposed a family of element-wise functions to lower the entropy and employ a learnable power function for simplicity and rescaling. Besides, we used convolution to alleviate the problem of degenerate solutions caused by the low-rank property of **SM** and introduced polarity-aware coefficient matrices to learn the complementary relationship between same-signed and opposite-signed values. We validated the effectiveness of the proposed PolaFormer in a series of vision tasks and additionally benchmarked on the LRA testbed to fairly compare with mainstream linear attention models. The experimental results demonstrated that our model has good compatibility with most attention-based models and measures up to a better balance between performance and efficiency.

Table 5: Comparison of classification results on the ImageNet-1K dataset. The default input resolution is $224^2$, except for the last row, which reports results for variants using a resolution of $384^2$.

| MODEL | RESO | PARAMS | FLOPS | ACC(%) |
|---|---|---|---|---|
| SBCFormer-B (Lu et al., 2024) | $224^2$ | 14M | 1.6G | 80.0 |
| SBCFormer-L (Lu et al., 2024) | $224^2$ | 19M | 2.7G | 81.1 |
| CAS-ViT-T (Zhang et al., 2024b) | $224^2$ | 22M | 3.5G | 82.3 |
| VisionMamba-T (Zhu et al., 2024) | $224^2$ | 7M | 1.1G | 76.1 |
| VisionMamba-S (Zhu et al., 2024) | $224^2$ | 26M | 3.7G | 80.6 |
| VisionMamba-B (Zhu et al., 2024) | $224^2$ | 98M | 13.7G | 81.9 |
| T2T-14 (Yuan et al., 2021a) | $224^2$ | 21.5M | 4.8G | 81.5 |
| T2T-19 (Yuan et al., 2021a) | $224^2$ | 39.2M | 8.5G | 81.9 |
| T2T-24 (Yuan et al., 2021a) | $224^2$ | 64.1M | 13.8G | 82.3 |
| CvT-13 (Wu et al., 2021) | $224^2$ | 20M | 4.5G | 81.6 |
| CvT-21 (Wu et al., 2021) | $224^2$ | 32M | 7.1G | 82.5 |
| CvT-13 (Wu et al., 2021) | $384^2$ | 20M | 16.3G | 83.0 |
| CvT-21 (Wu et al., 2021) | $384^2$ | 32M | 24.9G | 83.3 |
| HiViT-T (Zhang et al., 2023) | $224^2$ | 19M | 4.6G | 82.1 |
| HiViT-S (Zhang et al., 2023) | $224^2$ | 38M | 9.1G | 83.5 |
| HiViT-B (Zhang et al., 2023) | $224^2$ | 66M | 15.9G | 83.8 |
| PVT-T-FLatten (Han et al., 2023a) | $224^2$ | 11M | 1.9G | 77.8 |
| PVT-S-FLatten (Han et al., 2023a) | $224^2$ | 22M | 4.0G | 81.7 |
| PVTv2-b0-FLatten (Han et al., 2023a) | $224^2$ | 3.2M | 0.6G | 71.1 |
| PVTv2-b0-MobiAtt (Yao et al., 2024) | $224^2$ | 3.5M | 0.6G | 71.5 |
| PVTv2-b1-FLatten (Han et al., 2023a) | $224^2$ | 13M | 2.2G | 79.5 |
| Swin-S-FLatten (Han et al., 2023a) | $224^2$ | 51M | 8.7G | 83.5 |
| Swin-B-FLatten (Han et al., 2023a) | $224^2$ | 89M | 15.4G | 83.8 |
| PVT-T (Wang et al., 2021) | $224^2$ | 13M | 1.9G | 75.1 |
| PVT-T-PolaFormer | $224^2$ | 12M | 2.0G | $\textbf{78.8}_{+3.7}$ |
| PVT-S (Wang et al., 2021) | $224^2$ | 25M | 3.8G | 79.8 |
| PVT-S-PolaFormer | $224^2$ | 21M | 4.1G | $\textbf{81.9}_{+2.1}$ |
| PVTv2-b0 (Wang et al., 2022) | $224^2$ | 3.7M | 0.5G | 70.5 |
| PVTv2-b0-PolaFormer | $224^2$ | 3.4M | 0.6G | $\textbf{72.3}_{+1.8}$ |
| PVTv2-b1 (Wang et al., 2022) | $224^2$ | 13M | 2.1G | 78.7 |
| PVTv2-b1-PolaFormer | $224^2$ | 13M | 2.2G | $\textbf{80.2}_{+1.5}$ |
| Swin-S (Liu et al., 2021) | $224^2$ | 50M | 8.7G | 83.0 |
| Swin-S-PolaFormer | $224^2$ | 50M | 8.7G | $\textbf{83.6}_{+0.6}$ |
| Swin-B (Liu et al., 2021) | $224^2$ | 88M | 15.4G | 83.5 |
| Swin-B-PolaFormer | $224^2$ | 88M | 15.4G | $\textbf{83.8}_{+0.3}$ |
| Swin-S (Liu et al., 2021) | $384^2$ | 50M | 25.2G | 84.3 |
| Swin-B (Liu et al., 2021) | $384^2$ | 88M | 47.0G | 84.5 |
| Swin-S-PolaFormer | $384^2$ | 50M | 25.5G | $\textbf{84.7}_{+0.4}$ |

ACKNOWLEDGMENTS

This research is partially supported by National Natural Science Foundation of China (Grant No. 62372132), Shenzhen Science and Technology Program (Grant No. RCYX20221008092852077) and Australian Research Council (DE240100105, DP240101814, DP230101196). We would like to express our gratitude to Huawei funding and valuable discussions with Dr. Wei Zhou.

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

## A APPENDIX

This Appendix provides proof and supporting lemma for Theorem 1, followed by implementation details for various vision tasks. The source code is available in the supplementary material for reference.

- **Proof. A.1:** The mathematical proof and supporting lemmas of Theorem 1
- **Implementation Details. A.2:** Training settings for all experiments
- **Long Sequence Efficiency**
- **Comparison of the results of models with different G initializations**
- **Visualization of Attention Probability Distribution's Entropy**
- **Visualization of Attention Maps**

### A.1 PROOF OF THEOREM 1

**Theorem.** Let $\mathbf{x}, \mathbf{y}^n \in \mathbb{R}^d$ for $n = 1, \ldots N$, and dimensions are independently distributed. Given that $g : [0, +\infty) \mapsto [0, +\infty)$ is a differentiable function satisfying the condition $g'(x) > 0$ and $g''(x) > 0$ for all $x > 0$. Then, there exists such a function $g$ such that the PSE of the transformed sequence is strictly less than that of the original sequence. Specifically, we have:

$$\text{PSE}(\langle g(\mathbf{x}), g(\mathbf{y^1}) \rangle, \ldots, \langle g(\mathbf{x}), g(\mathbf{y}^N) \rangle) < \text{PSE}(\langle \mathbf{x}, \mathbf{y^1} \rangle, \ldots, \langle \mathbf{x}, \mathbf{y}^N \rangle). \tag{11}$$

**Proof.** We establish two lemmas to facilitate the proof of the main theorem.

**Lemma 1.** *Let $f$ be a function induced by $g : [0, +\infty) \mapsto [0, +\infty)$ with the conditions of $g'(x) > 0$ and $g''(x) > 0$, for all $x > 0$, defined as:*

$$f(\langle \mathbf{x}, \mathbf{y} \rangle) := \langle g(\mathbf{x}), g(\mathbf{y}) \rangle \tag{12}$$

*where $\mathbf{x}, \mathbf{y} \in \mathbb{R}^{d^+}, g(\mathbf{x}) = (g(x_1), \ldots, g(x_d))$. Then $f(x) > 0$, $f'(x) > 0$ and $f''(x) > 0$, for all $x \geq 0$.*

*Proof.* Consider the element-wise function $g$ for pairs of $(\mathbf{x}, \mathbf{y})$ with dimension $d$:

$$\begin{aligned} g(\mathbf{x}) &= (g(x_1), \ldots, g(x_d)) \\ g(\mathbf{y}) &= (g(y_1), \ldots, g(y_d)) \end{aligned} \tag{13}$$

Then, the inner-product between $g(\mathbf{x})$ and $g(\mathbf{y})$ is given by,

$$\langle g(\mathbf{x}), g(\mathbf{y}) \rangle = \sum_{i=1}^{d} g(x_i)g(y_i). \tag{14}$$

Because of the independence across dimensions, we apply Jensen's inequality, leveraging $g'(x) > 0$ and $g''(x) > 0$, yielding:

$$\begin{aligned} \mathbb{E}[f(\langle \mathbf{q}, \mathbf{k} \rangle)] = \mathbb{E}[\langle g(\mathbf{q}), g(\mathbf{k}) \rangle] &= \mathbb{E}[\sum_{i=1}^{d} g(q_i)g(k_i)] \\ &= \sum_{i=1}^{d} \mathbb{E}[g(q_i)g(k_i)] = \sum_{i=1}^{d} \mathbb{E}[g(q_i)]\mathbb{E}[g(k_i)] \\ &\leq \sum_{i=1}^{d} g(\mathbb{E}[q_i])g(\mathbb{E}[k_i]) \quad \text{(Jensen's Inequality)} \\ &= \langle g(\mathbb{E}[\mathbf{q}]), g(\mathbb{E}[\mathbf{k}]) \rangle = f(\langle \mathbb{E}[\mathbf{q}], \mathbb{E}[\mathbf{k}] \rangle) \\ &= f(\mathbb{E}[\langle \mathbf{q}, \mathbf{k} \rangle]) \end{aligned} \tag{15}$$

where $\mathbb{E}[\mathbf{q}] = (\mathbb{E}[q_1], \ldots, \mathbb{E}[q_d])$ denotes a vector. Consequently, we have the following results, *i.e.,*

$$\mathbb{E}[f(\langle \mathbf{q}, \mathbf{k} \rangle)] \leq f(\mathbb{E}[\langle \mathbf{q}, \mathbf{k} \rangle]), \tag{16}$$

indicating that $f$ is concave function having a positive second derivative. Also, according to the definition of $\mathbf{x}$ and $\mathbf{y}$, $f$ is obviously mapping from $[0, +\infty)$ to $[0, +\infty)$ with a positive first derivative.

$\square$

**Lemma 2.** *Given two positive values $(a, b)$, and function $f : [0, +\infty) \mapsto [0, +\infty)$ with the conditions of $f'(x) > 0$ and $f''(x) > 0$, we have $\mathrm{PSE}(f(a), f(b)) \leq \mathrm{PSE}(a, b)$.*

*Proof.* Consider the case $N = 2$ (extendable to $N > 2$). Without loss of generality, we assume $a > b, c := \frac{a}{b}$, then $c > 1$, and $\mathrm{PSE(a, b)}$ can be calculated as

$$
\begin{aligned}
H_1 &= -\left(\frac{a}{a+b} \log(\frac{a}{a+b}) + \frac{b}{a+b} \log(\frac{b}{a+b})\right) \\
&= -\left(\frac{c}{c+1} \log(\frac{c}{c+1}) + \frac{1}{c+1} \log(\frac{1}{c+1})\right) \\
&= \log(c+1) - \frac{c}{c+1} \log(c)
\end{aligned}
\tag{17}
$$

Then, we apply the kernel function $f$ on $(a, b)$, and it is mapped to $(f(a), f(b))$. Then, we define $d$ by $d := \frac{f(a)}{f(b)}$, and it is easy to prove that $d > c > 1$. Followed by Eq. (17), we can compute $\mathrm{PSE}(f(a), f(b))$ as:

$$
H_2 = \log(d+1) - \frac{d}{d+1} \log(d)
\tag{18}
$$

Through defining $h(x) = \log(x+1) - \frac{x}{x+1} \log(x), x > 1$, we have

$$
\begin{aligned}
h'(x) &= -\frac{\log(x)}{(x+1)^2} \\
h'(x) &\leq 0, \quad x > 1
\end{aligned}
\tag{19}
$$

indicating that $H_1 = h(c) > H_2 = h(c)$ for all $x > 1$, *i.e.,* $H_2 < H_1$. Therefore, all functions that satisfy the conditions have the effect of entropy decrease. $\square$

Now come back to the theorem. Firstly, we define $f$ induced by $g$ that

$$
f(\langle \mathbf{x}, \mathbf{y} \rangle) = \langle g(\mathbf{x}), g(\mathbf{y}) \rangle
\tag{20}
$$

From Lemma 1, we know that $f$ is a function with positive first and second derivative. Then by using Lemma 2, we have,

$$
\mathrm{PSE}(f(\langle \mathbf{x}, \mathbf{y}^1 \rangle), f(\langle \mathbf{x}, \mathbf{y}^2 \rangle)) < \mathrm{PSE}(\langle \mathbf{x}, \mathbf{y}^1 \rangle, \langle \mathbf{x}, \mathbf{y}^2 \rangle)
\tag{21}
$$

Therefore, the scaling effect can be achieved by the element-wise computation based on a function $g$ with positive first and second derivative. This allows for the removal of the softmax function, enabling linear complexity and lower entropy in the attention mechanism. $\square$

## A.2 IMPLEMENTATION DETAILS

**Classification.** In this task, we use the AdamW optimizer (Loshchilov & Hutter, 2019) to train all of our models for 400 epochs, including 20 epochs for linear warm-up. The basic learning rate is set to $1e-3$ for 1024 batch size. Additionally, we use a weight decay of $5e-2$. The training framework is developed on the top of the official Swin Transformer implementation made by Microsoft.

**Object Detection.** In this task, we utilize pretrained PVT models and Swin models on as the backbone and connect them to various detectors. Specifically, for the PVT model, we select from RetinaNet and Mask R-CNN as detectors, with the schedule set to $1\times$. For the Swin model, we choose the detector from Mask R-CNN and Cascade Mask R-CNN as detectors, where models using Mask R-CNN are experimented with under both $1\times$ and $3\times$ schedule settings, while models using Cascade Mask R-CNN case are trained under the $3\times$ schedule. All experiments follow the

`mmcv-detection` (Contributors, 2018) project. The training epoch is set to 12 per schedule and we use the AdamW optimizer with a learning rate of $1e-4$ and a weight decay of $1e-4$.

**Semantic Segmentation.** we employ pretrained PVT models and Swin models on two representative segmentation models, SemanticFPN and UperNet. The task is conducted based `mmcv-segmentation` (Contributors, 2018) project. The training interation is set to 40000 for PVT-SFPN models, 160000 for Swin-UperNet models by using AdamW optimizer with a learning rate of $2e-4$ and a weight decay of $1e-3$.

**Long Range Arena.** We evaluate the PolaFormer based on the official implementation of Skyformer (Chen et al., 2021). For Listops and Text Classification, we set batch size to 32 with $1e-4$ learning rate. For Pathfinder, we set batch size to 128 with $5e-4$ learning rate. For Image Classification, we set batch size to 256 with $1e-4$ learning rate. For Retrieval sub-task, we set batch size to 16 with $2e-4$ learning rate. All models are trained from scratch using the AdamW optimizer.

## A.3 LONG SEQUENCE EFFICIENCY

To evaluate the scalability of our model in such settings, we performed experiments on the Long-Range Arena (LRA) benchmark. These results demonstrate PolaFormer's efficiency and scalability for both high-resolution vision tasks and long-sequence NLP applications.

Table 6: Throughput and Peak Memory of various models. A denotes the accuracy, T denotes the throughput of each model and M denotes the peak memory cost.

| | | Softmax | Kernelized | Nystrom | Linformer | Informer | Skyformer | Pola(ours) |
|---|---|---|---|---|---|---|---|---|
| Img (1k) | A | 39.14 | 32.63 | 38.94 | 38.43 | 37.86 | 40.77 | 42.15 |
| | T | 736.36 | 862.32 | 1251.28 | 1613.19 | 85.85 | 923.04 | 1340.89 |
| | M | 9645 | 13013 | 5941 | 3471 | 5357 | 8091 | 4505 |
| Path (1k) | A | 70.39 | 69.86 | 69.34 | 65.39 | 56.44 | 70.73 | 70.53 |
| | T | 691.67 | 811.59 | 1125.08 | 1057.03 | 299.94 | 748.98 | 1065.63 |
| | M | 4831 | 6515 | 2980 | 1745 | 2687 | 4055 | 2286 |
| List (2k) | A | 38.71 | 38.46 | 37.95 | 36.44 | 37.05 | 38.69 | 37.35 |
| | T | 402.06 | 496.48 | 834.85 | 528.52 | 305.53 | 627.14 | 949.80 |
| | M | 4473 | 6084 | 1186 | 881 | 2737 | 1712 | 1151 |
| Text (4k) | A | 61.55 | 60.02 | 62.36 | 57.29 | 62.13 | 64.7 | 73.06 |
| | T | 252.06 | 327.27 | 1330.68 | 970.90 | 521.16 | 949.80 | 876.74 |
| | M | 17122 | 11720 | 2043 | 1742 | 5736 | 3082 | 1155 |
| Retri (4k) | A | 80.93 | 82.11 | 80.89 | 77.85 | 79.35 | 82.06 | 80.5 |
| | T | 116.30 | 144.83 | 496.48 | 424.18 | 142.94 | 348.60 | 344.93 |
| | M | 8947 | 10699 | 2011 | 1649 | 3399 | 2987 | 1139 |
| Avg | A | 58.14 | $56.62_{-1.52}$ | $57.90_{-0.24}$ | $55.08_{-3.06}$ | $54.57_{-3.57}$ | $59.39_{+1.25}$ | $60.72_{+2.58}$ |
| | T | 439.69 | $528.50_{\times1.20}$ | $1007.68_{\times2.29}$ | $918.77_{\times2.09}$ | $271.08_{\times0.62}$ | $719.51_{\times1.80}$ | $915.60_{\times2.08}$ |
| | M | 9003.6 | $9606.2_{\times1.07}$ | $2832.2_{\times0.31}$ | $1897.6_{\times0.21}$ | $3983.2_{\times0.44}$ | $3985.4_{\times0.44}$ | $2047.2_{\times0.22}$ |

## A.4 COMPARISON OF THE RESULTS WITH DIFFERENT INITIALIZATIONS OF COEFFICIENTS MATRICES

To assess the impact of $\mathbf{G}$ initialization on downstream tasks, we conducted additional experiments using five distinct initialization methods. These experiments were performed on a text classification (TEXT) task in Long Range Arena (LRA) with a sequence length of 4k, maintaining the same experimental setup as described in Table 4. The initialization strategies tested included Kaiming uniform, zero initialization, normal distribution ($\mathcal{N}(0,1)$), uniform distribution ($\mathcal{U}(0,1)$), and constant ones. The results are summarized in the table below:

| Init Comparison | Kaiming Uniform | Zeros | Normal(0,1) | Uniform(0,1) | Ones |
|---|---|---|---|---|---|
| Acc | 73.06 | 72.17 | 74.30 | 74.40 | 70.70 |

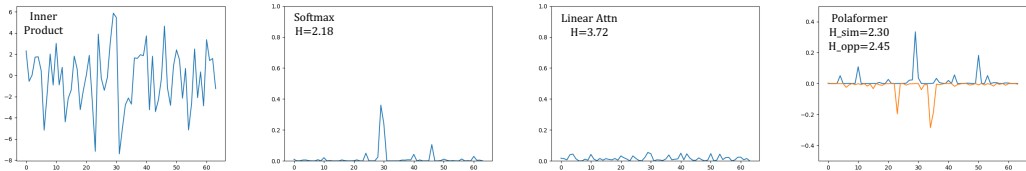

Figure 5: Visualization of different attention probability distributions

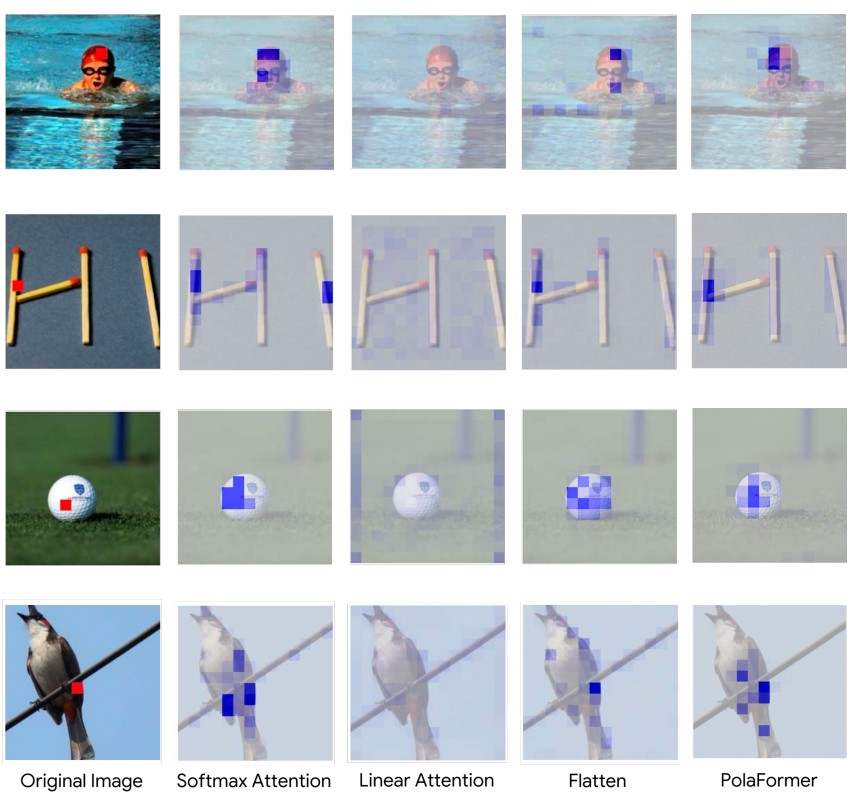

Figure 6: Visualization of the attention maps.

## A.5 VISUALIZATION OF ATTENTION PROBABILITY DISTRIBUTION'S ENTROPY

We compute the entropy of standard self-attention, linear attention (Katharopoulos et al., 2020) and PolaFormer. Additionally, we visualize the distribution of one row of the Attention Score matrix, as shown in Figure 5. It is clear that our model has a lower entropy than linear attention.

## A.6 VISUALIZATION OF ATTENTION MAPS

To further show the characteristics of accurate similarity calculation and low information entropy of our model. We visualize more examples shown in Figure 6. Thanks to the superiority of our designed kernel function, PolaFormer can calculate the similarity more accurately and focus on more relevant places.

