# OpenReview forum: "PolaFormer: Polarity-aware Linear Attention for Vision Transformers"
_ICLR.cc/2025/Conference — ICLR 2025 Poster_

### Official Review · Reviewer_EBEw · 2024-11-02

**Soundness:** 3
**Presentation:** 3
**Contribution:** 3
**Rating:** 6
**Confidence:** 5

**Summary:**

This paper aims to improve the state of the art linear transformers with linear complexity O(N) with N the sequence length. It focus on addressing two issues as identified: 1) Loss of negative values, and 2) loss of attention spikiness. The proposed method, PolaFormer, is based on the idea of separating the query-key pairs with their plarity into two branches, one for positive and one for negative. Other improvement includes making some previously hand-designed static parameters to be learnable. Experiments are conducted on image classification, object detection and segmentation, and the long range arena tasks.

**Strengths:**

Good and clear writing, with good background and context, detailed description of equations, and transformer mechnisam, and the literature review of previous linear transformers.

Visualisation is easy to understand

The key idea is delivered clearly, with the assistance of well designed charts and graphs.

The experiments show good margin as compared to previous alternative methods.

**Weaknesses:**

First point unclear to me is about the definition of negative values (Line 201). As clearly stated in softmax kernel function (Line 16), they operate in the output end of softmax function where it is deemed that all values are not negative. It is not about the input end of softmax where values can be either negative or positive. Under this context, I do not see anywhere negative values are lost or overlooked as kernel based linear attention is dealing with the softmax-ed space where no negative values exist at all.

Even if one wants to have negative values, what is the fundamental challenge with just changing the corresponding component, such as instead of using ReLU, one can simply use other activation functions allowing negative values such as leaky ReLU, Tanh etc. No discussion on this simple baseline choice is provided. This also implies that the motivation of going for more complex design is thus not strong. That being said, for being actionable, it is suggested that the authors compare their approach against baselines using activation functions that allow negative values, such as leaky ReLU or tanh or shift sigmoid with both negative and positive. This would help clarify the advantages of their more complex design over simpler alternatives.

It is hard to relate Eq (4) with the loss of negative values as discussed before such as Lines 190-194. Why ReLU based mapping will just operate on latter 3 terms and leave out the first term? This seems to be ad-hoc assumption. I would suggest that the authors provide a more detailed explanation or proof of how ReLU-based mappings specifically affect the terms in Eq (4), and why this leads to information loss.

The other learnable parts to replacing static hand-designed values is good bit but hard to be claimed as addressing major challenges in this context.

As this work is about scalability of transformer along with the length of input sequences, what is the range of each experiment. In general, the sequences are not that long for vision tasks like image classification, object detection and segmentation. They are not the best suitable test applications. LRA may give longer sequences which is thus better for test. I would suggest the authors to provide specific sequence lengths for each experiment. Additionally, including experiments on tasks with longer sequences would be helpful to better demonstrate the scalability of this approach.

Also, the scalability of the proposed model along with the increase of sequence length is not evaluated, which should be a key aspect for this problem. Including an experiment or analysis that explicitly shows how this model's performance and efficiency scale with increasing sequence length, compared to baseline methods, would be important and insightful.

Similarly, the ablation study is best done with long sequence based tasks, but not short ones like image-net classification.

Table 4: while $\alpha$ in Eq (9) is learnable, why in this table it looks like the value is set to 3/5/7? Or I misunderstand? Besides, please show what are the learned value of $\alpha$ for each other experiment.

**Questions:**

Please check the weakness parts

---

> ### Author Response · Authors · 2024-11-20
> **Response to Reviewer EBEw (1)**
>
> Thank you for reviewing our paper! Before addressing your comments point by point, we would like to re-clarify the some key concepts involved this work to provide a comprehensive context for our explanations:
>
> - **Linear Attention**: Existing linear attention approaches typically **replace** the softmax operation over the query-key dot product (softmax$(qk^\top)$) with a carefully designed kernel function $\phi$. This substitution allows the computation of qkv to be *reordered*, achieving linear complexity and improved efficiency.
> - **Non-Negativity Constraint and Information Loss**: To ensure non-negativity (as the output of softmax is inherently non-negative), these kernel functions often rely on mappings like ReLU that sets specific negative dimensions of $\mathbf{q}$ and $\mathbf{k}$ to zero (detailed examples refer to following replies). This introduces significant deviations from the original query-key dot products and results in partial information loss, which we will discuss in greater detail in subsequent responses.
> - **Our Contributions**. The information loss caused by strict non-negativity constraints is commonly overlooked in prior works, leading to higher entropy and reduced discriminative power in the resulting attention mechanism. Our contributions address these challenges through several innovations:
>     1. We introduce a more relaxed framework that separates query-key interactions into same-signed and opposite-signed flows, recovering lost information.
>     2. We design a learnable matrix to compensate for the non-negativity constraint by effectively mimicking the subtractive operation between flows.
>     3. We incorporate a learnable power-scaling mechanism to control positive sequence entropy, maintaining spikiness and precision.
>     4. This relaxed framework is particularly beneficial for vision tasks, such as object localization and segmentation, that demand high precision in query-key interactions.
>
> We hope these clarifications provide a solid foundation for understanding our approach. We will delve deeper into the specifics in our responses to your comments. Thank you once again for your constructive feedback, and please feel free to share any follow-up questions—we are more than happy to discuss any aspect of the work further!

---

> ### Author Response · Authors · 2024-11-20
> **Response to Reviewer EBEw (2)**
>
> >**Weakness**: First point unclear to me is about the definition of negative values (Line 201). As clearly stated in softmax kernel function (Line 16), they operate in the output end of softmax function where it is deemed that all values are not negative. It is not about the input end of softmax where values can be either negative or positive. Under this context, I do not see anywhere negative values are lost or overlooked as kernel based linear attention is dealing with the softmax-ed space where no negative values exist at all.
> >
> >It is hard to relate Eq (4) with the loss of negative values as discussed before such as Lines 190-194. Why ReLU based mapping will just operate on latter 3 terms and leave out the first term? This seems to be ad-hoc assumption. I would suggest that the authors provide a more detailed explanation or proof of how ReLU-based mappings specifically affect the terms in Eq (4), and why this leads to information loss.
> >
> >The other learnable parts to replacing static hand-designed values is good bit but hard to be claimed as addressing major challenges in this context.
>
> Thanks for the question. To address your concerns, we will clarify the definitions and provide detailed reasoning for how PolaFormer handles the issues related to negative values and information loss:
>
> **Non-Negative Restrictions in Attention Mechanisms.** In standard self-attention, the output is defined as $\operatorname{output}=\mathbf{SV}$, where $\mathbf{S}=\operatorname{softmax}(\mathbf{QK}^\top)$. The softmax function ensures that all elements of $\mathbf{S}$ are non-negative, and each output token becomes a probability-weighted sum of $\mathbf{V}$. While there are no constraints on the signs of individual elements in $\mathbf{QK}^\top$, applying the exponential function in softmax naturally converts all values to positive. As noted in prior works ([1], [2]), a critical property of $\mathbf{S}$ is that all its elements must be non-negative.
>
> In linear attention, where $\mathbf{S} = \phi(\mathbf{Q}) \phi(\mathbf{K})^\top$, non-negativity must be enforced explicitly by the kernel function $\phi$. If this condition is not met, the presence of negative elements in $\mathbf{q}_i$ or $\mathbf{k}_j$ can lead to negative values in $\mathbf{q}_i \mathbf{k}_j^\top$, causing instability in training (e.g., exploding gradients).
>
> **The Loss in Existing Linear Attention Methods.** To meet the non-negativity requirement, existing linear attention methods often map *negative values* in $\mathbf{q}$ and $\mathbf{k}$ to *zero*.  However, these approaches overlook the large-magnitude negative values in $\mathbf{q}$ or $\mathbf{k}$. During the inner product computation, the ignored dimensions (where $\mathbf{q}^{-}$ and $\mathbf{k}^{-}$ are zero or close to zero) result in inaccurate $\langle \mathbf{q, k} \rangle$ calculations. As shown in the following formula, only the positive dimensions contribute to the computation, while the dimensions involving negative values are ignored. This severely limits the representational power of the model.
>
> \begin{equation}
> \begin{aligned}
>     \langle \mathbf{q},\mathbf{k} \rangle &=
>     \langle \mathbf{q}^{+},\mathbf{k}^{+} \rangle +
>     \langle \mathbf{q}^{+},\mathbf{k}^{-} \rangle +
>     \langle \mathbf{q}^{-},\mathbf{k}^{+} \rangle +
>     \langle \mathbf{q}^{-},\mathbf{k}^{-} \rangle\\\\
>     &=
>     \langle \mathbf{q}^{+},\mathbf{k}^{+} \rangle +
>     \langle \mathbf{q}^{+},0 \rangle +
>     \langle 0,\mathbf{k}^{+} \rangle +
>     \langle 0,0 \rangle \\\\
>     &=\langle \mathbf{q}^{+},\mathbf{k}^{+}\rangle
> \end{aligned}
> \end{equation}
>
> **How PolaFormer Addresses These Challenges**. Our algorithm aims to include all four components in the first line of the equation above. To achieve this, we compute these four components separately. Using the ReLU function, we extract $q^{+}$, $q^{-}$, $k^{+}$, and $k^{-}$, ensuring that $\mathbf{S} = \phi(\mathbf{Q})\phi(\mathbf{K})^\top$ contains only non-negative elements while simultaneously preserving the information from all dimensions of the $\mathbf{qk}$ vectors.
>
> [1] Zhen Qin, Weixuan Sun, Hui Deng, Dongxu Li, Yunshen Wei, Baohong Lv, Junjie Yan, Lingpeng Kong, and Yiran Zhong. Cosformer: Rethinking softmax in attention. In The Tenth International Conference on Learning Representations (ICLR). OpenReview.net, 2022.
>
> [2] Angelos Katharopoulos, Apoorv Vyas, Nikolaos Pappas, and Fran¸cois Fleuret. Transformers are rnns: Fast autoregressive transformers with linear attention. In International conference on machine learning, pp. 5156–5165. PMLR, 2020.

---

> > ### Author Response · Authors · 2024-11-20
> > **Response to Reviewer EBEw (3)**
> >
> > **An Illustrative Example.** Here we provide a toy example to demonstrate why our method contains the full information in inner-product:
> >
> > \begin{aligned}
> >     q&=[-1,-3,3,2,-5,3]\\\\
> >     k&=[2 ,-5,-2,5,4,3]
> > \end{aligned}
> >
> > then, to the standard inner-product, we compute $\langle q,k \rangle$ by
> > \begin{equation}
> > \begin{aligned}
> > \langle q,k \rangle = \sum_{i=1}^{6}q_ik_i &= (-2) + (15)+(-6)+(10)+(-20)+(9)\\\\
> > &= \underbrace{(15 + 10 + 9)} _ {\text{same-signed flow}} -\underbrace{(2+6+20)} _ {\text{opposite-signed flow}}
> > \end{aligned}
> > \end{equation}
> > In existing linear attention method, they have to keep all elements of $q,k$ non-negative. No matter what kind of kernel function, inevitably, negative values are mapped to 0 or an infinitesimal positive number. Here, we use ReLU as the example. Then, the inner-product in linear attention is computed as followed:
> > \begin{aligned}
> > \operatorname{ReLU}(q)&=[0,0,3,2,0,3] \\\\
> > \operatorname{ReLU}(k)&=[2,0,0,5,4,3] \\\\
> > \langle \operatorname{ReLU}(q),\operatorname{ReLU}(k) \rangle &= \sum_{i=1}^{6}q_ik_i \\\\
> > &= (0) + (0)+(0)+(10)+(0)+(9)\\\\
> > &=\underbrace{(0 + 10 + 0)} _ {\text{same-signed flow}}-\underbrace{(0 + 0 + 0)} _ {\text{opposite-signed flow}}
> > \end{aligned}
> > Here, contributions from the opposite flow are completely lost, which should contribute to the inner product.
> >
> > In response to our PolaFormer, we use a learnable coefficiency matrixes $G^s,G^o$ to balance both similar flow and opposite flow. Here, we abstract the coefficiency matrixes as a function $f$,
> >
> > we have
> > \begin{aligned}
> > q_1&=[\operatorname{ReLU}(q);\operatorname{ReLU}(-q)]=[0,0,3,2,0,3;1,3,0,0,5,0] \\\\
> > q_2&=[\operatorname{ReLU}(q);\operatorname{ReLU}(-q)]=[1,3,0,0,5,0;0,0,3,2,0,3] \\\\
> > k&=[\operatorname{ReLU}(k),\operatorname{ReLU}(-k)]=[2,0,0,5,4,3;0,5,2,0,0,0]\\\\
> > \end{aligned}
> > \begin{aligned}
> > \langle q,k \rangle&=(\underbrace{\langle q^{+},k^{+} \rangle +
> > \langle q^{-},k^{-} \rangle} _ {\text{similar flow}}) -
> > (\underbrace{\langle q^{+},k^{-} \rangle +
> > \langle q^{-},k^{-} \rangle} _ {\text{opposite flow}})\\\\
> > &=\langle q_1,k \rangle - \langle q_2,k \rangle\\\\
> > &\approx f(\langle q_1,k \rangle , \langle q_2,k \rangle) \\\\
> > &=f(15+10+9,2+6+20)
> > \end{aligned}
> >
> > It is clear that all six terms are preserved in PolaFormer.

---

> ### Author Response · Authors · 2024-11-20
> **Response to Reviewer EBEw (4)**
>
> >**Weakness**: Even if one wants to have negative values, what is the fundamental challenge with just changing the corresponding component, such as instead of using ReLU, one can simply use other activation functions allowing negative values such as leaky ReLU, Tanh etc. No discussion on this simple baseline choice is provided. This also implies that the motivation of going for more complex design is thus not strong. That being said, for being actionable, it is suggested that the authors compare their approach against baselines using activation functions that allow negative values, such as leaky ReLU or tanh or shift sigmoid with both negative and positive. This would help clarify the advantages of their more complex design over simpler alternatives.
>
> Thank you for raising this point. As stated in Line 190 of our paper, each element of the $\mathbf{S}$ matrix must be strictly non-negative because it **directly** relies on the dot products of kernel maps. Allowing negative values in the $\mathbf{S}$ matrix violates this requirement, leading to instability during training. Empirically, when negative values were permitted, we observed gradient explosions, causing the training process to fail entirely. To validate this, we conducted experiments on the Long-Range Arena (LRA) benchmark while allowing negative values in $\mathbf{S}$. However, all attempts resulted in task failure, with the loss diverging to NaN.
>
> >**Weakness**: Table 4: while $\alpha$ in Eq (9) is learnable, why in this table it looks like the value is set to 3/5/7? Or I misunderstand? Besides, please show what are the learned value of $\alpha$ for each other experiment.
>
> Thank you for your question. To clarify, $\alpha$ in our learnable power function is a hyperparameter that controls the magnitude range of the outputs, which are regularized by the sigmoid function in the range (0, 1). Specifically, the power function is defined as:
> $\mathbf{P} = 1 + \alpha \cdot \operatorname{sigmoid}(w_1, \dots, w_d)$
> where only the parameters $w_i$ are learnable. The "+1" term ensures that the power function satisfies the first- and second-order derivative requirements specified in Theorem 1. This design ensures both stability and smoothness in the learning process.
>
> In Table 4, we compare model performance using fixed values of $\alpha$ set to 3, 5, and 7, respectively, to illustrate the effect of different magnitude ranges on downstream tasks. We sincerely apologize for the typo and have corrected it in the revised version. Thank you for bringing it to our attention!

---

> ### Author Response · Authors · 2024-11-20
> **Response to Reviewer EBEw (5)**
>
> >**Weakness**: As this work is about scalability of transformer along with the length of input sequences, what is the range of each experiment. In general, the sequences are not that long for vision tasks like image classification, object detection and segmentation. They are not the best suitable test applications. LRA may give longer sequences which is thus better for test. I would suggest the authors to provide specific sequence lengths for each experiment. Additionally, including experiments on tasks with longer sequences would be helpful to better demonstrate the scalability of this approach. Also, the scalability of the proposed model along with the increase of sequence length is not evaluated, which should be a key aspect for this problem. Including an experiment or analysis that explicitly shows how this model's performance and efficiency scale with increasing sequence length, compared to baseline methods, would be important and insightful.
>
> Thank you for your insightful question and suggestion. Our **primary focus** is on improving the accuracy of query-key interactions in efficient vision transformers, which is crucial for tasks like detection and segmentation that require precise localization. These tasks often rely on token-to-token correlations, making accurate similarity computations critical.
>
> **Scalability in Vision Tasks.** While scalability with longer sequences or higher resolutions is not the primary objective, we conducted experiments with varying image resolutions to evaluate the scalability of our approach. Specifically, we tested resolutions ranging from $224^2$ to $384^2$ as reported in Lines 535–537 and Table 6 of the paper. To address your suggestion, we have supplemented these experiments with additional results at a resolution of $512^2$. The results demonstrate that our algorithm maintains consistent advantages as the resolution increases. The reason we did not initially explore even higher resolutions was to ensure a fair comparison with existing works. However, the additional experiments on $512^2$ images further validate the robustness of our method for higher-resolution inputs.
>
> **Scalability in NLP Tasks.** NLP tasks inherently involve longer sequences compared to vision tasks. To evaluate the scalability of our model in such settings, we performed experiments on the Long-Range Arena (LRA) benchmark. In the revised version of the paper, we have explicitly highlighted the sequence lengths used in different LRA sub-tasks to address this concern.
> |  |  |  |  |  |  |  |  |  |  |  |  |  |  |  |  |  |  |  |
> |---|---|---|---|---|---|---|---|---|---|---|---|---|---|---|---|---|---|---|
> | **TASK** | Image\(1k\) |  |  | Pathfinder\(1k\) |  |  | Listops\(2k\) |  |  | Text\(4k\) |  |  | Retrieval\(4k\) |  |  | Avg |  |  |
> | **MODEL** | Acc | Throughput | Memory | Acc | Throughput | Memory | Acc | Throughput | Memory | Acc | Throughput | Memory | Acc | Throughput | Memory | Acc | Throughput | Memory |
> | Softmax | 39\.14 | 736\.3595506 | 9645 | 70\.39 | 691\.6728232 | 4831 | 38\.71 | 402\.0613497 | 4473 | 61\.55 | 252\.0615385 | 17122 | 80\.93 | 116\.3016859 | 8947 | 58\.144 | 439\.6913896 | 9003\.6 |
> | Kernelized | 32\.63 | 862\.3157895 | 13013 | 69\.86 | 811\.5913313 | 6515 | 38\.46 | 496\.4848485 | 6084 | 60\.02 | 327\.2709114 | 11720 | 82\.11 | 144\.8309392 | 10699 | 56\.616 | 528\.498764 | 9606\.2 |
> | Nystrom | 38\.94 | 1251\.28401 | 5941 | 69\.34 | 1125\.081545 | 2980 | 37\.95 | 834\.8535032 | 1186 | 62\.36 | 1330\.680203 | 2043 | 80\.89 | 496\.4848485 | 2011 | 57\.896 | 1007\.676822 | 2832\.2 |
> | Linformer | 38\.43 | 1613\.193846 | 3471 | 65\.39 | 1057\.032258 | 1745 | 36\.44 | 528\.516129 | 881 | 57\.29 | 970\.9037037 | 1742 | 77\.85 | 424\.1812298 | 1649 | 55\.08 | 918\.7654333 | 1897\.6 |
> | Informer | 37\.86 | 85\.85033568 | 5357 | 56\.44 | 299\.9359268 | 2687 | 37\.05 | 305\.5291375 | 2737 | 62\.13 | 521\.1610338 | 5736 | 79\.35 | 142\.9356598 | 3399 | 54\.566 | 271\.0824187 | 3983\.2 |
> | Skyformer | 40\.77 | 923\.0422535 | 8091 | 70\.73 | 748\.9828571 | 4055 | 38\.69 | 627\.138756 | 1712 | 64\.7 | 949\.7971014 | 3082 | 82\.06 | 348\.5957447 | 2987 | 59\.39 | 719\.5113426 | 3985\.4 |
> | PolaFormer | 42\.15 | 1340\.890026 | 4505 | 70\.53 | 1065\.626016 | 2286 | 37\.35 | 949\.7971014 | 1151 | 73\.06 | 876\.735786 | 1155 | 80\.5 | 344\.9263158 | 1139 | 60\.718 | 915\.595049 | 2047\.2 |
>
> These results highlight the efficiency of our method in terms of both time and memory consumption for longer sequences, reinforcing the advantages of our linear attention mechanism.

---

> > ### Author Response · Authors · 2024-11-20
> > **Response to Reviewer EBEw (6)**
> >
> > **Focus Differences Between Vision-oriented and NLP-oriented Linear Attention.** Our design is particularly tailored for vision transformers, where accurate query-key correlations are paramount. The demands and focus for vision and NLP tasks differ in the following ways:
> >
> > - Vision Tasks: These tasks prioritize accurate query-key interactions and require precise computation of similarities between tokens to achieve accurate localization and segmentation.
> > - NLP Tasks: In contrast, NLP tasks emphasize the relationship between local and global information. Handling long sequences effectively is critical for capturing global context.
> >
> > Given these differences, our method excels in vision tasks by accurately recovering dot-products between query and key vectors. This also explains why our approach achieves significant improvements in tasks like segmentation and detection (by 4.6%). While our model demonstrates versatility across both domains, its design inherently aligns better with the requirements of vision transformers.
> >
> > Finally, we created a table comparing similarity measures to provide a clearer understanding of our algorithm. Notably, the "Attention Score" row highlights that $\mathbf{q}_i$ and $\mathbf{k}_i$ represent the $i$-th elements of the $\mathbf{qk}$ vectors, with the summation symbol indicating the dimensions of the $\mathbf{qk}$ vectors involved in the calculation (the rest are set to zero). It can be observed that only the standard Transformer and our algorithm include all dimensions, whereas other approaches lose some components (corresponding to the last three terms of Eq (4). This ultimately explains why our algorithm achieves the highest accuracy in visualization.
> >
> > | Similarity Comparison  | Vanilla Transformer |Linear Attention|FLatten Transformer|PolaFormer|
> > |---------|----------|----------|----------|----------|
> > | Attentionn Score (before normalized)| exp($\sum_iq_ik_i$)|$\sum_{q_i,k_i\geq0}q_ik_i$|$\sum_{q_i,k_i\geq0}q_i^pk_i^p$|$f(\sum_{q_ik_i\geq0}q_ik_i,\sum_{q_ik_i\leq0}q_ik_i)$|
> > | Complexity     |  $O(N^2)$    | $O(N)$ |$O(N)$ |$O(N)$ |
> > | Information Integrity |  $\checkmark$| |  |$\checkmark$|
> > | Positive Score   |  $\checkmark$| $\checkmark$ |  $\checkmark$ |$\checkmark$|
> > | Low Entropy  |  $\checkmark$| |  $\checkmark$ |$\checkmark$|

---

> > > ### Comment · Reviewer_EBEw · 2024-11-24
> > > **post rebuttal interaction**
> > >
> > > Thank the authors for providing additional explanation, details, and evaluations which are very helpful, making this work easier to understand. All these information would be useful to be part of the paper and appendix. Overall I think this is a good piece of work given in-depth analysis and stronger performance. I will raise the rating accordingly.
> > >
> > > From the tables provided in the response, it is also interesting to note that PolarFormer works even more accurately than softmax in some cases. Originally, I thought softmax would be the upper bound of these linear attention models, but the experiments indicates otherwise. This gives rise to some questions:
> > > 1) Does this negativity issue is also with the softmex attention
> > > 2) Any aspects with PolarFormer that go beyond the negativity, have not been realized? The table on similarity comparison cannot reflect this if any, as  PlolarFormer and Softmax vanilla both have the same set of features except different complexity. I feel that a bit closer look into this will be critical, as the current perspective is mostly constrained on linear attention scope.

---

> > > > ### Author Response · Authors · 2024-11-25
> > > > **Response to Reviewer EBEw (7)**
> > > >
> > > > Thank you for your swift response and your further questions. All additional information you mentioned will be added into the appendix after discussion!
> > > >
> > > > >**Question0**: I thought softmax would be the upper bound of these linear attention models, but the experiments indicates otherwise.
> > > >
> > > > This is really a good question. It is worth noting that in the exisiting literature on linear attention such as Skyformer [1] and Hedgehog [2] have already outperformed vanilla softmax attention in average accuracy on the LRA benchmark. In line with our design, our approach achieves a more consistent improvement on vision tasks (e.g., classification on Image-1K (LRA Benchmark), detection on MS-COCO and segmentation on ADE20K).
> > > >
> > > > **Rationale beyind the improvement.** Regarding the improvement of linear attention over softmax-based attention, prior works [1-2], to the best of our knowlegde, have not provided detailed explanations. Based on our empirical  observations, we hypothesize that the performance gains could stem from multiple factors, including:
> > > >
> > > > 1) Reduced Overparameterization: linear models may generalize better due to simpler parameteriztion;
> > > > 2) Softmax sometimes can cause trouble in some cases as it is:
> > > >     a) too spiky: the exponential scaling in softmax can overly amplify certain parts of the sequence, in extreme cases (one element in one vector with a super large value), almost all weaker signials will be potentially ignored.
> > > >     b) not flexible: due to the fixed function $exp(qk^\top)$ in softmax, the contributions of* different dimensions* of the vector cannot be flexibly distinguished or re-scaled.
> > > >
> > > > While these hypotheses offer plausible explanations, rigorous theoretical proofs and empirical validations remain unexplored in this work and this is the reason we did not provide too much discussions in the original submission. We sincerely agree with reviewer that this could highlight an important direction for future exploration.
> > > >
> > > > [1] Yifan Chen, Qi Zeng, Heng Ji, and Yun Yang. Skyformer: Remodel self-attention with gaussian kernel and nystrom method. In Marc’Aurelio Ranzato, Alina Beygelzimer, Yann N. Dauphin, Percy Liang, and Jennifer Wortman Vaughan (eds.), Advances in Neural Information Processing Systems (NeurIPS), pp. 2122–2135, 2021.
> > > >
> > > > [2] Zhang, Michael, et al. "The Hedgehog & the Porcupine: Expressive Linear Attentions with Softmax Mimicry." The Twelfth International Conference on Learning Representations.
> > > >
> > > > >**Question1**: Does this negativity issue is also with the Softmax attention
> > > >
> > > > Thank you for your question. The "negativity issue" we refer to is *specific to* linear attention and does not occur with vanilla softmax attention. This is because softmax attention computes the full inner product (preserving all terms) before applying the softmax operation to reduce entropy.
> > > >
> > > > In contrast, linear attention often omits negative terms in its computation, as these values are mapped to zero (kindly refer to our earlier responses for details). Therefore, softmax attention does not exhibit negativity issues. However, it can face challenges in certain tasks due to its normalization behavior, which constrains outputs to the range (0,1). This normalization can reduce the discriminative power of the model in scenarios requiring a broader or more subtle range of values.
> > > >
> > > > >**Question2**:  Any aspects with PolarFormer that go beyond the negativity, have not been realized?
> > > >
> > > > Thank you for this important question. While the primary focus of our paper is addressing negativity and spikiness in attention mechanisms, yet during our experiments, we found the following aspects could be further improved:
> > > >
> > > > 1. We observed that mixed-precision training can occasionally lead to numerical instability and the current design may not fully support highly efficient parallelization. To address these challenges, we plan to investigate engineering tricks in online softmax [1] and Flash Attention [2] to enhance the model's scalability, stability, and compatibility with fp16 precision training.
> > > >
> > > > 2. Linear attention mechanisms are theoretically less efficient in decoder-only models (e.g., GPT, Bloom, LLaMA) due to the reliance on causal masking $((QK)\odot M)V$, which prevents the prioritization of KV calculation. Existing implementations often use architectures proposed in [3] or adapt to bi-directional structures like BERT. However, there is significant room to enhance their universality and adaptability to a broader range of LLM architectures.
> > > >
> > > > [1] Milakov, Maxim, and Natalia Gimelshein. "Online normalizer calculation for softmax." arXiv preprint arXiv:1805.02867 (2018).
> > > >
> > > > [2] Shah, Jay, et al. "FlashAttention-3: Fast and Accurate Attention with Asynchrony and Low-precision." The Thirty-eighth Annual Conference on Neural Information Processing Systems.
> > > >
> > > > [3] Yang, Songlin, et al. "Gated Linear Attention Transformers with Hardware-Efficient Training." Forty-first International Conference on Machine Learning.

---

> > > > ### Author Response · Authors · 2024-11-25
> > > > **Response to Reviewer EBEw (8)**
> > > >
> > > > >**Question3**:  The table on similarity comparison cannot reflect this if any, as PlolarFormer and Softmax vanilla both have the same set of features except different complexity. I feel that a bit closer look into this will be critical, as the current perspective is mostly constrained on linear attention scope.
> > > >
> > > > Thank you for your insightful comment. We would like to clarify that the previous table we provided compares the calculation of attention scores *before normalization*, and the softmax-based attention and linear attention differ in feature sets (our features are concatenated along the channel dimension) and normalization methods (Softmax uses the inner products of qk for joint normalization, while our method normalizes q and k separately). To make these distinctions clearer, we have updated the table to include additional rows:
> > > >
> > > > | Similarity Comparison  | Vanilla Transformer | PolaFormer|
> > > > |---------|----------|----------|
> > > > | Attentionn Score (before normalized)| exp($\sum_iq_ik_i$)|$f(\sum_{q_ik_i\geq0}q_ik_i,\sum_{q_ik_i\leq0}q_ik_i)$|
> > > > | Complexity     |  $O(N^2)$   |$O(N)$ |
> > > > | Positive Score   |  $\checkmark$| $\checkmark$|
> > > > | Norm factor | $\sum_j{exp(qk_j^\top)}$ | $\phi(q)\sum_j{(\phi(k_j)^\top}$|
> > > > | Scaling for each dimension| Equally | Learnable|
> > > >
> > > > We agree with the reviewer that activation function design is a critical aspect. As previously discussed, softmax itself may be suboptimal in certain scenarios. Exploring alternative *learnable* activations, such as KAN [1], could indeed represent a promising future direction.
> > > >
> > > > [1] Liu, Ziming, et al. "Kan: Kolmogorov-arnold networks." arXiv preprint arXiv:2404.19756 (2024).
> > > >
> > > > We hope the above discussion has clarified your questions. Should you have any additional queries or require further elaboration, we are more than happy to address them! If our explanations meet your expectations, we would sincerely appreciate your consideration in updating the ratings, as your recognition of our work means a lot to us!

---

> > > > > ### Comment · Reviewer_EBEw · 2024-11-25
> > > > > **Post-rebuttal Discussion - R2**
> > > > >
> > > > > Thank the authors for the following reply on these new questions. All these would be useful with added clarity for this work, thus being worthwhile to be part of it. It is fair and justified for my rating increase given the current form.

---

> > > > > > ### Author Response · Authors · 2024-11-25
> > > > > > **Thank You for Reading and Consideration**
> > > > > >
> > > > > > Dear Reviewer EBEw，
> > > > > >
> > > > > > Thank you so much for your thoughtful advice and positive response. We are delighted that our reply addressed all of your questions and concerns, and we greatly appreciate you raising your score.
> > > > > >
> > > > > > Your helpful comments and suggestions have strengthened our draft, and we believe they have significantly enhanced the overall quality of our work.
> > > > > >
> > > > > > Once again, we would like to express our sincere gratitude for your consistent support and attention.
> > > > > >
> > > > > > Best, \
> > > > > > Authors of Submission 2313.

---

### Official Review · Reviewer_veYX · 2024-11-04

**Soundness:** 3
**Presentation:** 3
**Contribution:** 3
**Rating:** 6
**Confidence:** 4

**Summary:**

This paper introduces PolaFormer, a new attention mechanism designed for vision transformers that aims to improve both expressiveness and efficiency. Traditional linear attention methods reduce computational complexity by approximating the softmax function but often lose critical information by ignoring negative query-key interactions and producing less discriminative attention maps with higher entropy. PolaFormer tackles this issue by explicitly modeling both positive and negative query-key interactions, ensuring a more comprehensive capture of relational information. Additionally, it employs a learnable power function to adjust the attention distribution's entropy, restoring the sharpness characteristic of softmax-based attention. The authors back their approach with theoretical analysis and demonstrate its effectiveness through extensive experiments across various vision tasks, showing notable performance improvements while maintaining linear computational complexity.

**Strengths:**

1. The key strength is the introduction of polarity-aware attention. By decomposing the query and key vectors into positive and negative components (as in Equation (3)), they capture all possible interactions: positive-positive, negative-negative, positive-negative, and negative-positive. This is a significant departure from traditional methods that only consider positive interactions due to non-negative feature maps.
2. They provide solid theoretical analysis. For example, in Theorem 1, they prove that using a function \( g \) with positive first and second derivatives (like their learnable power function) reduces the entropy of the attention distribution. This mathematical proof gives credibility to their claim that their method restores the "spikiness" of the attention weights.
3. The experimental results are impressive. On ImageNet-1K classification, their PolaFormer variants outperform the baselines by significant margins. For instance, DeiT-T-PolaFormer improves Top-1 accuracy by up to 6.3% over other DeiT variants. In object detection tasks on COCO, they achieve improvements ranging from 2.3% to 4.6% in AP scores. These aren't just marginal gains—they're substantial improvements that demonstrate the practical value of their method.
4. Despite the added complexity of handling negative interactions and using a learnable power function, they manage to keep the computational complexity linear with respect to sequence length \( N \), as shown in their complexity analysis (Equation (10)). They also report faster inference speeds compared to other models with similar FLOPs, which is a big deal for real-world applications.

**Weaknesses:**

1. Introducing a learnable power function and additional parameters like the polarity coefficients \( G_s \) and \( G_o \) could introduce training challenges, such as sensitivity to initialization or convergence issues. The paper doesn't discuss whether they encountered any of these problems or how they addressed them.
2. Since attention mechanisms are also fundamental in NLP, it would have been interesting to see PolaFormer's performance on language tasks. The paper focuses solely on vision tasks, so we don't know if the benefits carry over to NLP applications.
3. While they do perform an ablation study (as shown in Table 3), it could be more comprehensive. For instance, they could explore how different values of the scaling factor in the learnable power function affect performance, or how sensitive the model is to the choice of convolutional modules used to increase the rank of the attention map.

**Questions:**

1. Did introducing the learnable power function and the polarity coefficients \( G_s \) and \( G_o \) affect training stability? For example, did you encounter issues like vanishing or exploding gradients? If so, how did you mitigate them?
2. How sensitive is the model's performance to the initialization and learning of the polarity coefficients \( G_s \) and \( G_o \)? Did you notice any significant performance drops or instability when varying these parameters?
3. Have you tried applying PolaFormer to NLP tasks like machine translation or language modeling? Since attention is crucial in these areas too, it would be interesting to see if your method provides benefits there.
4. You theoretically show that your method reduces entropy in the attention distribution. Did you measure the entropy empirically in your experiments to confirm this? It would be interesting to see a plot or some data showing how the entropy changes with your method compared to others.
5. The attention weight visualizations in Figure 1 are helpful. Could you provide more examples, perhaps showing how PolaFormer focuses on relevant parts of the input in different tasks or layers? This could provide more intuitive insight into how your method works.

---

> ### Author Response · Authors · 2024-11-20
> **Response to Reviewer veYX (1)**
>
> > **Weakness 1**: Introducing a learnable power function and additional parameters like the polarity coefficients $\mathbf{G}_s$ and $\mathbf{G}_o$ could introduce training challenges, such as sensitivity to initialization or convergence issues. The paper doesn't discuss whether they encountered any of these problems or how they addressed them.
> >&
> > **Question1**: Did introducing the learnable power function and the polarity coefficients $\mathbf{G}_s$ and $\mathbf{G}_o$ affect training stability? For example, did you encounter issues like vanishing or exploding gradients? If so, how did you mitigate them?
> > &
> > **Question2**: How sensitive is the model's performance to the initialization and learning of the polarity coefficients $\mathbf{G}_s$ and $\mathbf{G}_o$? Did you notice any significant performance drops or instability when varying these parameters?
>
> **Training Stability.** We appreciate your insightful questions regarding the training stability and sensitivity of the model to the learnable power function and polarity coefficients, $\mathbf{G}_s$ and $\mathbf{G}_o$. Below, we address each aspect in detail: During our experiments, we initialized $\mathbf{G}$ using the Kaiming uniform initialization by default. We did not encounter any issues related to gradient vanishing or exploding, and the training process remained stable throughout. To further evaluate the robustness of the model, we conducted additional experiments to investigate the effect of different initialization strategies on training dynamics and downstream performance.
>
> **Sensitivity to Initialization.** To assess the impact of $\mathbf{G}$ initialization on downstream tasks, we conducted additional experiments using five distinct initialization methods. These experiments were performed on a text classification (TEXT) task in Long Range Arena (LRA) with a sequence length of 4k, maintaining the same experimental setup as described in Table 4 (Line 449). The initialization strategies tested included Kaiming uniform, zero initialization, normal distribution ($\mathcal{N}(0, 1)$), uniform distribution ($\mathcal{U}(0, 1)$), and constant ones. The results are summarized in the table below:
>
> | Init Comparison| Kaiming Uniform |Zeros|Normal(0,1)|Uniform(0,1)|Ones|
> |---------|----------|----------|----------|----------|----------|
> | Acc|  73.06 |72.17|74.30|74.40|70.70|
>
> These results show that while initialization strategies have varying effects on performance, our algorithm maintains its overall advantage compared to baseline methods. Notably, uniform initialization performed slightly better, suggesting it is a strong alternative to Kaiming uniform. However, none of the initialization methods led to significant instability or convergence issues.
>
> > **Weakness2**: Since attention mechanisms are also fundamental in NLP, it would have been interesting to see PolaFormer's performance on language tasks. The paper focuses solely on vision tasks, so we don't know if the benefits carry over to NLP applications.
> > &
> > **Question3**：Have you tried applying PolaFormer to NLP tasks like machine translation or language modeling? Since attention is crucial in these areas too, it would be interesting to see if your method provides benefits there.
>
> Thank you for your thoughtful comment. To demonstrate the versatility of PolaFormer, we have validated its performance on the Long-Range Arena (LRA) benchmark, which includes NLP tasks such as **TEXT, LISTOPS, and RETRIEVAL** (Line 432-452). As shown in **Table 4**, PolaFormer achieves results that are either better or comparable to state-of-the-art methods on these NLP tasks, highlighting its potential for broader applicability.
>
> Our decision to prioritize vision tasks stems from the fact that efficient vision transformer architectures are predominantly benchmarked under vision settings. This ensures a fair comparison and thorough evaluation within the established scope. Nevertheless, we acknowledge that further validation on large-scale NLP tasks would provide a more comprehensive assessment of PolaFormer’s capabilities.
>
> >**Weakness3**: While they do perform an ablation study (as shown in Table 3), it could be more comprehensive. For instance, they could explore how different values of the scaling factor in the learnable power function affect performance, or how sensitive the model is to the choice of convolutional modules used to increase the rank of the attention map.
>
> Thanks. We have already conducted ablation studies on the learnable power parameter $\alpha$ in Table 4 (Lines 449 to 452). Additionally, experiments on the choice of convolution models are presented in Table 3, where we compared deformable convolutions and depth-wise convolutions. The results indicate that DWC is more suitable for our model and downstream tasks, providing better performance in these scenarios.

---

> ### Author Response · Authors · 2024-11-20
> **Response to Reviewer veYX (2)**
>
> >**Quesstion4**:You theoretically show that your method reduces entropy in the attention distribution. Did you measure the entropy empirically in your experiments to confirm this? It would be interesting to see a plot or some data showing how the entropy changes with your method compared to others.
>
> Thank you for your question. To address this question, we compute the entropy of standard self-attention, linear attention and PolaFormer. Additionally, we visualize the distribution of one row of the Attention Score matrix in the Appendix A.5.
>
> | Method  | Softmax | Linear | PolaFormer (sim,opp) |
> |---------|---------|--------|----------------------|
> | **Entropy** | 2.18    | 3.72   | ( 2.30, 2.45 )          |
>
> >**Question5**: The attention weight visualizations in Figure 1 are helpful. Could you provide more examples, perhaps showing how PolaFormer focuses on relevant parts of the input in different tasks or layers? This could provide more intuitive insight into how your method works.
>
> Thank you for your question. This will help us to further show the characteristics of accurate similarity calculation and low information entropy of our model. We visualize more examples as fig1 in the Appendix A.6

---

### Official Review · Reviewer_KD6g · 2024-11-06

**Soundness:** 3
**Presentation:** 3
**Contribution:** 3
**Rating:** 8
**Confidence:** 4

**Summary:**

To overcome the shortcomings of current linearized self-attention, the paper proposes polarityaware linear attention mechanism that attends to both both same-signed and opposite-signed query-key interactions. the latter on is oftern ignored by existing methods. Besides, the proposed polarity-aware attention can also addresses the loss of attention spikeness.

**Strengths:**

1. The paper is well written and organized.
2. The paper provides solid motivation for the method and the proposed approach is well justified.
3. The experimental results show promising performance.

**Weaknesses:**

1. After reading the paper, I am still not sure how the non-negativity constraint is perserved. Especially when a learnable matrix $G$ is applied. Could authors provide more explanation on how the learnable matrix $G$ can perserve the non-negativity constraint?

2. Some latest baselines are missing in the paper. For instance, authors should consider incorporating the latest work [1] for comparsion. This baseline also proposes a new linear self-attention to achieve both high expressiveness capacity and low computation complexity.  The comparison on image classification or object detection would be helpful for a comprehensive assessment of the propsoed method.

3. One minor error in line 98. Negative-negative should be Negative-positive.

[1] Han, Dongchen, et al. "Agent attention: On the integration of softmax and linear attention." ECCV 2024.

**Questions:**

See weakness

---

> ### Author Response · Authors · 2024-11-20
> **Response to Reviewer KD6g (1)**
>
> > **Weakness1**: After reading the paper, I am still not sure how the non-negativity constraint is perserved. Especially when a learnable matrix $\mathbf{G}$ is applied. Could authors provide more explanation on how the learnable matrix $\mathbf{G}$ can perserve the non-negativity constraint?
>
> Thank you for your attention to the polarity-aware coefficients matrix and your thoughtful question.
>
> **1. Strict non-negativity constraint and its limitations.** In existing works, ensuring the rigorous non-negativity of the dot product between $\mathbf{q}$ and $\mathbf{k}$ often relies on kernel functions such as ReLU or ELU+1 due to the unpredictability of $\mathbf{q}$ and $\mathbf{k}$ during training. Specifically, these kernel functions, regardless of their design, can lead to significant drawbacks. For example, ReLU maps all negative values to zero, resulting in **substantial dimensional loss** during dot product calculations. This *reduces* the contribution of certain dimensions to the similarity, leading to *diminished spikiness* or *discriminativeness* in the representation. Alternatively, functions like abs distort the relationship between similarity and opposition, making it *difficult to distinguish* between aligned and opposing directions in the vector space. Such limitations hinder accurate similarity calculations, which are particularly critical for *downstream vision tasks* requiring precise discriminative features.
>
> **2. Relaxed non-negativity constraint.** To address these challenges, we adopt a more flexible strategy aimed at minimizing information loss. Instead of enforcing a strict non-negativity constraint, we separate the same-signed and opposite-signed relationships across dimensions, modeling them independently and subsequently concatenating them for joint learning. This approach allows us to *relax* the non-negativity constraint, enabling the model to *preserve* more information and improve *query-key interaction accuracy*.
>
> **3. Learnable Conpensation.** The learnable matrix $\mathbf{G}$ plays a pivotal role in **compensating** for the relaxed non-negativity constraint. By capturing and leveraging the inverse relationships between the two flows (same-signed and opposite-signed), $\mathbf{G}$ facilitates better integration of information. Empirically, our experiments and visualizations show that $\mathbf{G}$ successfully **learns** the opposing relationships during training, thereby somewhat ensuring non-negativeness in a way that **aligns** with the dot product calculation. This design not only enhances the information flow between the two channels but also significantly improves overall performance in downstream tasks.

---

> ### Author Response · Authors · 2024-11-20
> **Response to Reviewer KD6g (2)**
>
> >**Weakness2**: Some latest baselines are missing in the paper. For instance, authors should consider incorporating the latest work [1] for comparsion. This baseline also proposes a new linear self-attention to achieve both high expressiveness capacity and low computation complexity. The comparison on image classification or object detection would be helpful for a comprehensive assessment of the propsoed method.
>
> Thank you for your constructive feedback. We appreciate your suggestion to incorporate [1]. However, there are significant considerations that make direct comparison with [1] less appropriate in our case:
>
> **Why Agent Attention is an unfair comparison.** The method proposed in [1] uses a novel agent attention mechanism to achieve a high expressiveness capacity. As discussed in the Related Work section (Line 145), agent attention introduces agent tokens $\mathbf{A}$ that aggregate and broadcast global information effectively. This method leverages the Softmax properties while incorporating elements of linear attention. However,
> 1. Its complexity is **not** strictly linear with respect to sequence length. Agent tokens in [1] scale with the sequence length as the model size increases. While they are fewer in number than the queries, their scaling nature deviates from the linear complexity characteristic of our approach. This makes a direct comparison between the two methods less equitable.
> 2. Furthermore, [1] does **not** address the **softmax-free problem**, which is a core focus of linear attention approach. Since [1] retains the Softmax mechanism for key operations, it diverges from the primary goals of linear attention methodologies.
>
> **Our comparions with SOTA (2024) works.** In our work, we aimed to ensure a fair and meaningful comparison by including baselines that align more closely with the objectives of linear attention. As shown in Table 5, we benchmarked our method against four state-of-the-art works (Mobile Attention (ICML'24) [2], CAS-ViT (Arxiv'24) [3], Vision Mamba (ICML'24) [4], and SBCformer (WACV'24) [5]) from 2024. These models were selected for their relevance and adherence to the principles of efficiency and scalability, providing a robust framework for evaluating the strengths of our approach.
>
> [1] Han, Dongchen, et al. "Agent attention: On the integration of softmax and linear attention." ECCV 2024.
>
> [2] Zhiyu Yao, Jian Wang, Haixu Wu, Jingdong Wang, and Mingsheng Long. Mobile attention: Mobilefriendly linear-attention for vision transformers. In Forty-first International Conference on Machine Learning (ICML), 2024.
>
> [3] Tianfang Zhang, Lei Li, Yang Zhou, Wentao Liu, Chen Qian, and Xiangyang Ji. Cas-vit: Convolutional additive self-attention vision transformers for efficient mobile applications. CoRR, abs/2408.03703, 2024b.
>
> [4] Lianghui Zhu, Bencheng Liao, Qian Zhang, Xinlong Wang, Wenyu Liu, and Xinggang Wang. Vision mamba: Efficient visual representation learning with bidirectional state space model. In Forty-first International Conference on Machine Learning (ICML), 2024.
>
> [5] Xiangyong Lu, Masanori Suganuma, and Takayuki Okatani. Sbcformer: Lightweight network capable of full-size imagenet classification at 1 FPS on single board computers. In IEEE/CVF Winter Conference on Applications of Computer Vision (WACV), pp. 1112–1122. IEEE, 2024.
>
> >**Weakness3**: One minor error in line 98. Negative-negative should be Negative-positive.
>
> Thank you for your carefully reading. We have corrected this error in the revised draft.

---

### Official Review · Reviewer_KHpw · 2024-11-06

**Soundness:** 3
**Presentation:** 3
**Contribution:** 3
**Rating:** 8
**Confidence:** 3

**Summary:**

Linear attention replaces the softmax over query-key dot products with a kernel function that makes the attention operation linear in kernel space, which breaks the quadratic complexity complexity in the sequence length.
This paper proposes a variant of linear attention where (1) the positive and negative components of the query and key vectors are separated and (2) the individual components are made more "peaky" by applying a per-component $x \leadsto x^p$ transformation where the powers $p>1$ are learned.

**Strengths:**

S1.  The method presented in this paper is simple and well justified

S2. Paper generally easy to read

S3. results outperform the full attention baseline (I merely expected it to be a good approximation)

**Weaknesses:**

W1. The main dawback of the experiments is that there is not experiment on truly high resolution image, where this method would fully benefit from the linear complexity

W2. Some parts are not clear (see below)


Unclear points:
- it would be useful to specify from the intro that the method is applied with full training -- it is unclear until the experiments that this is not fine-tuning and not a drop-in replacement for softmax attention at inference time

L133: better = more accurate or faster?

L162: relationship between d and D ?

L248 it is unclear how G^s and G^o can be trained since they depend on the batch size N -- or is N assumed to be fixed?

eq (7) maybe not necessary to define what entropy is... Also the argument about better accounting for negative correlation is repeated 4-6x in the paper. By shaving these repetitions you could make room to move the proof to the main paper.

L300 IIUC d' = d since g() does just a pointwise mapping -- please mention this explicitly

Maybe some useful related work on the power kernels : [Tolias et al, Particular Object Retrieval With Integral Max-Pooling of CNN Activations, ICLR'16]

**Questions:**

please clarify how G^s and G^o can be trained

---

> ### Author Response · Authors · 2024-11-20
> **Response to Reviewer KHpw (1)**
>
> > Weakness 1: The main dawback of the experiments is that there is not experiment on truly high resolution image, where this method would fully benefit from the linear complexity
>
> Thank you for highlighting this important aspect. To ensure a **fair** comparison with existing efficient vision transformer architectures, we initially trained the proposed PolaFormer model using the ImageNet-1K dataset under standard resolution ($224 \times 224$) and a higher resolution ($384 \times 384$), as reported in Lines 535–537 and Table 5 of the paper. To address your concern, we extended our evaluation to an even higher resolution ($512 \times 512$), testing both the baseline and PolaFormer under these more demanding conditions. The results, summarized in the table below, show that PolaFormer consistently maintains its advantage, demonstrating robust performance even at high resolutions.
>
> | Model  | Resolution |Accuracy|
> |---------|----------|----------|
> | Swin-S    |  $384^2$   |84.3 |
> | Swin-B     |  $384^2$    | 84.5|
> | Swin-S-PolaFormer |  $384^2$   |84.7|
> | Swin-S   |  $512^2$   | 84.6|
> | Swin-S-PolaFormer  | $512^2$ |85.0|
>
> **Disscussion.** Regarding the significance of high-resolution tests, it is worth noting that the key challenge in *vision tasks* such as detection and segmentation is not solely the resolution itself but the **accuracy** of query-key interactions. Effective activation of the query vector for corresponding key vectors ensures that the objects of interest can be captured. PolaFormer’s design, which focuses on recovering accurate relationships in dot products and minimizing information loss, aligns with this aim and enhances performance across resolutions.
>
> **Additional Validation: Long Sequence Efficiency.** To evaluate the scalability of our model in such settings, we performed experiments on the Long-Range Arena (LRA) benchmark. In the revised version of the paper, we have explicitly highlighted the sequence lengths used in different LRA sub-tasks to address this concern.
>
> | **TASK** | Image(1k) |  |  | Pathfinder(1k) |  |  | Listops(2k) |  |  | Text(4k) |  |  | Retrieval(4k) |  |  | Avg |  |  |
> |---|---|---|---|---|---|---|---|---|---|---|---|---|---|---|---|---|---|---|
> | **MODEL** | **Acc** | **Throughput** | **Memory** | **Acc** | **Throughput** | **Memory** | **Acc** | **Throughput** | **Memory** | **Acc** | **Throughput** | **Memory** | **Acc** | **Throughput** | **Memory** | **Acc** | **Throughput** | **Memory** |
> | Softmax | 39\.14 | 736\.3595506 | 9645 | 70\.39 | 691\.6728232 | 4831 | 38\.71 | 402\.0613497 | 4473 | 61\.55 | 252\.0615385 | 17122 | 80\.93 | 116\.3016859 | 8947 | 58\.144 | 439\.6913896 | 9003\.6 |
> | Kernelized | 32\.63 | 862\.3157895 | 13013 | 69\.86 | 811\.5913313 | 6515 | 38\.46 | 496\.4848485 | 6084 | 60\.02 | 327\.2709114 | 11720 | 82\.11 | 144\.8309392 | 10699 | 56\.616 | 528\.498764 | 9606\.2 |
> | Nystrom | 38\.94 | 1251\.28401 | 5941 | 69\.34 | 1125\.081545 | 2980 | 37\.95 | 834\.8535032 | 1186 | 62\.36 | 1330\.680203 | 2043 | 80\.89 | 496\.4848485 | 2011 | 57\.896 | 1007\.676822 | 2832\.2 |
> | Linformer | 38\.43 | 1613\.193846 | 3471 | 65\.39 | 1057\.032258 | 1745 | 36\.44 | 528\.516129 | 881 | 57\.29 | 970\.9037037 | 1742 | 77\.85 | 424\.1812298 | 1649 | 55\.08 | 918\.7654333 | 1897\.6 |
> | Informer | 37\.86 | 85\.85033568 | 5357 | 56\.44 | 299\.9359268 | 2687 | 37\.05 | 305\.5291375 | 2737 | 62\.13 | 521\.1610338 | 5736 | 79\.35 | 142\.9356598 | 3399 | 54\.566 | 271\.0824187 | 3983\.2 |
> | Skyformer | 40\.77 | 923\.0422535 | 8091 | 70\.73 | 748\.9828571 | 4055 | 38\.69 | 627\.138756 | 1712 | 64\.7 | 949\.7971014 | 3082 | 82\.06 | 348\.5957447 | 2987 | 59\.39 | 719\.5113426 | 3985\.4 |
> | PolaFormer | 42\.15 | 1340\.890026 | 4505 | 70\.53 | 1065\.626016 | 2286 | 37\.35 | 949\.7971014 | 1151 | 73\.06 | 876\.735786 | 1155 | 80\.5 | 344\.9263158 | 1139 | 60\.718 | 915\.595049 | 2047\.2 |
>
> These results demonstrate PolaFormer’s efficiency and scalability for both high-resolution vision tasks and long-sequence NLP applications. We appreciate your feedback, which allowed us to highlight the robustness and versatility of our approach in more challenging scenarios. The results will be updated correspondingly in the revised manuscript.

---

> ### Author Response · Authors · 2024-11-20
> **Response to Reviewer KHpw (2)**
>
> >**Weakness2.1**: It would be useful to specify from the intro that the method is applied with full training -- it is unclear until the experiments that this is not fine-tuning and not a drop-in replacement for softmax attention at inference time
>
> Thanks for your constructive suggestion. We acknowledge that the current version of the manuscript could better highlight the training setup of our method. To clarify:
> 1. Classification Tasks: The proposed PolaFormer model was trained from scratch on the ImageNet-1K dataset, as described in Section 5.1 (Line 355).
> 2. Detection and Segmentation Tasks: For downstream tasks such as detection and segmentation, PolaFormer was fine-tuned following the pretraining stage, as detailed in Sections 5.2 and 5.3 (Lines 377 and 414).
>
> We appreciate your feedback and will revise the main body in future versions to explicitly outline these details upfront. Additionally, we will include further explanations in the supplementary material to clearly distinguish the training setups.
>
> >**Weakness2.2**: L133: better = more accurate or faster?
>
> Thank you for pointing out the ambiguity in our summary of related work on efficient vision transformers. In this context, "better" specifically refers to *accuracy*. While many efficient vision transformer architectures successfully reduce computational complexity, their accuracy often falls short of standard transformers. We will revise this sentence to clearly articulate that the trade-off in existing methods is between reduced complexity and maintaining competitive accuracy.
>
> >**Weakness2.3**: L162: relationship between d and D?
>
> Thank you for your question. To clarify, $D$ represents the dimensionality of the input vectors, while $d$ is the dimensionality of each attention head. In multi-head self-attention, the total input dimensionality $D$ is divided equally among $h$ attention heads. Thus, the relationship is $D = h \times d$. We will update the text to explicitly define this relationship for clarity.
>
> >**Weakness2.5**: Eq (7) maybe not necessary to define what entropy is... Also the argument about better accounting for negative correlation is repeated 4-6x in the paper. By shaving these repetitions you could make room to move the proof to the main paper.
>
> Thank you for your question. Each row of $\phi(\mathbf{Q})\phi(\mathbf{K})^\top$ is not normalized, in other words, it's not a probability distribution, and there is no such property as information entropy. Thus, we defined The Positive Sequence Entropy(PSE) in Eq(7) to use a unified information entropy calculation for all sequences without probability normalization.
>
> >**Weakness2.6**:L300 IIUC d' = d since g() does just a pointwise mapping -- please mention this explicitly
>
> Thank you for your carefully reading. We have mentioned "$d' = d$ since $g()$ does just a pointwise mapping" explicitly in the revised draft.
>
> >**Weakness2.7**:Maybe some useful related work on the power kernels : [Tolias et al, Particular Object Retrieval With Integral Max-Pooling of CNN Activations, ICLR'16]
>
> Thank you for your advice. We have incorporated the discussion of this work into the Related Work section in the revised draft.

---

> ### Author Response · Authors · 2024-11-20
> **Response to Reviewer KHpw (3)**
>
> >**Weakness2.4**: L248 it is unclear how $\mathbf{G}^s$ and $\mathbf{G^o}$ can be trained since they depend on the batch size $N$ or is $N$ assumed to be fixed?
> >\&
> >**Question**: please clarify how G^s and G^o can be trained
>
> Thank you for your question. The matrices $\mathbf{G}^s$ and $\mathbf{G}^o$ are derived as part of the linear transformations applied to the input $\mathbf{X}$, in a manner similar to the generation of the $\mathbf{Q}$, $\mathbf{K}$, and $\mathbf{V}$. Thus, The batch size $N$ of $\mathbf{G}$ matches the batch size of $\mathbf{X}$.
> Below, we outline the detailed process:
> 1. **Input Transformation**:
>    The input matrix $\mathbf{X}$ (of shape $N \times L\times D$, where $N$ is the batch size, $L$ is the sequence length and $D$ is the feature dimension) is passed through a linear layer that maps $D$ to $4D$ to produce $\mathbf{Q}$, $\mathbf{K}$, $\mathbf{V}$, and $\mathbf{G}$:
>
>     \begin{aligned}
>         &\mathbf{Q,K,V,G} = \operatorname{Linear}(D,4D)(\mathbf{X})
>     \end{aligned}
>
> 2. **Splitting $\mathbf{G}$**:
>    The matrix $\mathbf{G}$ (of shape $N \times L \times d$ for each head) is split into two equal parts, $\mathbf{G}^s$ and $\mathbf{G}^o$, each with shape $N \times L \times \frac{d}{2}$:
>     \begin{aligned}
>    \text{Concat}([\mathbf{G}^s, \mathbf{G}^o]) = \mathbf{G}
>        \end{aligned}
>
> 3. **Attention Flows**:
>    The attention mechanism operates in two flows:
>    - **Same-signed Flow** ($\mathbf{O}^s$): Computed based on $\mathbf{Q}$ and $\mathbf{K}$ with the same sign.
>    - **Opposite-signed Flow** ($\mathbf{O}^o$): Computed based on $\mathbf{Q}$ and $\mathbf{K}$ with opposite signs.
>
>    Each flow produces outputs $\mathbf{O}^s$ and $\mathbf{O}^o$, which match the shapes of $\mathbf{V}^s$ and $\mathbf{V}^o$:
>     \begin{aligned}
>     \mathbf{O}^s, \mathbf{O}^o = \text{Attention}(\mathbf{Q}, \mathbf{K}, \mathbf{V})
>        \end{aligned}
>
> 4. **Element-wise Fusion with $\mathbf{G}^s$ and $\mathbf{G}^o$**:
>    The outputs $\mathbf{O}^s$ and $\mathbf{O}^o$ are fused with $\mathbf{G}^s$ and $\mathbf{G}^o$ through a Hadamard product ($\odot$):
>     \begin{aligned}
> \mathbf{Z}^s = \mathbf{G}^s \odot \mathbf{O}^s, \mathbf{Z}^o = \mathbf{G}^o \odot \mathbf{O}^o
>     \end{aligned}
>
> 5. **Concatenation and Multi-Head Fusion**:
>    Finally, the results of the two flows ($\mathbf{Z}^s$ and $\mathbf{Z}^o$) are concatenated and combined across all attention heads to produce the final output:
>    \begin{equation}
>     \begin{aligned}\mathbf{Z} = \text{Concat}([\mathbf{Z}^s, \mathbf{Z}^o]) \quad \text{(per head)}
>     \end{aligned}
> \end{equation}
>
>
> These operations, including the linear transformation and attention computations, are differentiable and fully trainable within the standard backpropagation. Training $\mathbf{G}^s$ and $\mathbf{G}^o$ is therefore achieved jointly with the rest of the network parameters during end-to-end training. We will incorporate this detailed explanation into the supplementary material to provide additional clarity. Thank you for bringing this to our attention.

---

> > ### Comment · Reviewer_KHpw · 2024-11-26
> >
> > Thanks for the explanations.
> >
> > I read the other reviews and the explanations. I think the paper in its current state is suitable for publication.

---

### Author Response · Authors · 2024-11-21
**A Summary of Reviews**

Thank you to all reviewers for taking the time to review our paper and providing valuable and constructive comments! We are very grateful to all reviewers for giving our paper recognition and in the following aspects:

1) The method is a **good approximation** (reviewer KHpw)
2) The paper provides **solid motivation** for the method and the proposed approach is **well justified**. (reviewer KD6g)
3) This paper provides solid **theoretical analysis** and **mathematical proof,** which give credibility to the method.(reviewer veYX)
4) Good visualisation, well-designed charts (reviewer EBEw)
5) Good writing and easy to read (all reviewers)
6) The experimental results are **impressive** and show good margin as compared to previous alternative methods.(all reviewers)

Once again, I extend my heartfelt thanks to all the reviewers for your invaluable feedback on our paper!
We supplemented this paper with a few additional experiments and works, including：
1) Image classification tasks at higher resolutions.
2) Throughput and peak memory cost in LRA tasks.
3) Compute and visualize the entropy of attention score after mapped by different functions, as shown in Appendix 5.
4) Plot more examples of attention maps, as shown in Appendix 6.

You can find our individual responses below your review comments. **If you have any more concerns or questions, we are entirely open to continuing the discussion with you!**

---

### Meta-Review · Area_Chair_Vfyr · 2024-12-23

**Metareview:**

This paper proposes a new linear attention mechanism that fully leverages the positive and negative components of the query/key vectors in computing attention maps by separately learning query-key interactions according to their polarities. The authors provide theoretical analysis and demonstrate its effectiveness through extensive experiments across various tasks, showing notable performance improvements while maintaining linear computational complexity.
All reviewers appreciated the clear motivation, novelty, and comprehensive analysis. The main concerns raised by reviewers were unclear exposition, missing experiments/analyses, and a lack of comparisons with recent methods. The authors’ detailed rebuttal addressed most of these concerns, resulting in unanimous acceptance at the end of the discussion. AC thus recommends acceptance.

**Additional Comments On Reviewer Discussion:**

The main concerns raised by reviewers were unclear exposition, missing experiments/analyses, and a lack of comparisons with recent methods. The authors’ point-to-point rebuttal addressed most of these concerns so that no significant issues remain.

---

### Decision · Program_Chairs · 2025-01-22

Accept (Poster)